# Division of labour between PP2A-B56 isoforms at the centromere and kinetochore

**Giulia Vallardi, Lindsey A Allan, Lisa Crozier, Adrian T Saurin\***

Division of Cellular Medicine, School of Medicine, University of Dundee, Dundee, United Kingdom

**Abstract** PP2A-B56 is a serine/threonine phosphatase complex that regulates several major mitotic processes, including sister chromatid cohesion, kinetochore-microtubule attachment and the spindle assembly checkpoint. We show here that these key functions are divided between different B56 isoforms that localise to either the centromere or kinetochore. The centromeric isoforms rely on a specific interaction with Sgo2, whereas the kinetochore isoforms bind preferentially to BubR1 and other proteins containing an LxxIxE motif. In addition to these selective binding partners, Sgo1 helps to anchor PP2A-B56 at both locations: it collaborates with BubR1 to maintain B56 at the kinetochore and it helps to preserve the Sgo2/B56 complex at the centromere. A series of chimaeras were generated to map the critical region in B56 down to a small C-terminal loop that regulates the key interactions and defines B56 localisation. Together, this study describes how different PP2A-B56 complexes utilise isoform-specific interactions to control distinct processes during mitosis.

DOI: https://doi.org/10.7554/eLife.42619.001

**\*For correspondence:**
a.saurin@dundee.ac.uk

**Competing interests:** The authors declare that no competing interests exist.

## Introduction

Protein Phosphatase 2A (PP2A) is a major class of serine/threonine phosphatase that is composed of a catalytic (C), scaffold (A) and regulatory (B) subunit. Substrate specificity is mediated by the regulatory B subunits, which can be subdivided into four structurally distinct families: B (B55), B' (B56), B' (PR72) and B'' (Striatin) (*Seshacharyulu et al., 2013*).

In humans, the B subunits are encoded by a total of 15 separate genes which give rise to at least 26 different transcripts and splice variants; therefore, each of the four B subfamilies are composed of multiple different isoforms (*Seshacharyulu et al., 2013*). Although these isoforms are thought to have evolved to enhance PP2A specificity, there is still no direct evidence that isoforms of the same subfamily can regulate specific pathways or processes. Perhaps the best indirect evidence that they can comes from the observation that B56 isoforms localise differently during mitosis (*Bastos et al., 2014*; *Nijenhuis et al., 2014*). However, even in these cases, it is still unclear how this differential localisation is achieved or why it is needed.

We addressed this problem by focussing on prometaphase, a stage in mitosis when PP2A activity is essential to regulate sister chromatid cohesion (*Kitajima et al., 2006*; *Riedel et al., 2006*; *Tang et al., 2006*), kinetochore-microtubule attachments (*Foley et al., 2011*; *Kruse et al., 2013*; *Suijkerbuijk et al., 2012*; *Xu et al., 2013*) and the spindle assembly checkpoint (*Espert et al., 2014*; *Nijenhuis et al., 2014*). Crucially, all of these mitotic functions are controlled by PP2A-B56 complexes that localise to either the centromere or the kinetochore.

The kinetochore is a multiprotein complex that assembles on centromeres to allow their physical attachment to microtubules. This attachment process is stochastic and error-prone, and therefore it is safeguarded by two key regulatory processes: the spindle assembly checkpoint (SAC) and

**eLife digest** The cells in our body are a hive of activity, but that activity must be kept under control. This is never more critical than when a cell divides, because unchecked cell division can lead to cancer. Fortunately, enzymes called kinases and phosphatases exist to control the countless proteins in a cell; these enzymes help ensure that each step of cell division is complete before moving on to the next.

Kinases control other proteins by adding bulky phosphate groups to them, while phosphatases remove those groups. For a long time, phosphatases were assumed to be less specific than their kinase counterparts. Yet it has now become clear that phosphatases achieve specificity by interacting with a range of regulatory subunits.

A phosphatase called PP2A oversees a number of key steps in cell division by working together with its regulatory B56 subunit. In human cells, there are five separate genes that encode B56 subunits, and all of these B56 'isoforms' were thought to exert the same influence on the PP2A phosphatase. The fact, however, that different isoforms are found at different locations within the cell suggested otherwise.

To investigate this, Vallardi et al. focused on a particular stage of cell division when the activity of the PP2A-B56 complex is essential. Before a cell divides it duplicates its genetic material and the two copies of each chromosome are held together until the cell is ready to pull them apart. The experiments compared two representative B56 isoforms: one that concentrates at the centromere, the region where the copied chromosomes are held together; and another found at the kinetochore, a nearby structure that is involved in pulling the two chromosomes apart. By eliminating all but one isoform and measuring the ensuing activity of the PP2A-B56 complex, Vallardi et al. could differentiate between the main regulatory roles of each isoform. These experiments showed that B56 isoforms control separate processes during cell division, which mirrors their different locations within the cell.

Next, Vallardi et al. looked at the receptor proteins that recruit each isoform to its position. Removing or relocating different receptors showed how they anchor select B56 isoforms in different positions while the associated PP2A enzymes get to work on different processes. Further experiments using 'hybrid' subunits made from parts of two different B56 isoforms then helped to reveal the site on the B56 subunits that determines which receptors they bind to. Together these findings show that slight differences between each B56 isoform ultimately dictate where they localise and what processes they control when cells divide.

DOI: https://doi.org/10.7554/eLife.42619.002

kinetochore-microtubule error-correction. The SAC preserves the mitotic state until all kinetochores have been correctly attached to microtubules, whereas the error-correction machinery removes any faulty microtubule attachments that may form. The kinase Aurora B is critical for both processes because it phosphorylates the kinetochore-microtubule interface to destabilise incorrectly attached microtubules and it reinforces the SAC, in part by antagonising Knl1-PP1, a kinetochore phosphatase complex needed for SAC silencing (*Saurin, 2018*). These two principal functions of Aurora B are antagonised by PP2A-B56, which localises to the Knl1 complex at the outer kinetochore by binding directly to BubR1 (*Foley et al., 2011*; *Kruse et al., 2013*; *Suijkerbuijk et al., 2012*; *Xu et al., 2013*). This interaction is mediated by the B56 subunit, which interacts with a phosphorylated LxxIxE motif within the kinetochore attachment regulatory domain (KARD) of BubR1 (*Wang et al., 2016a*; *Wang et al., 2016b*).

As well as localising to the outer kinetochore, PP2A-B56 also localises to the centromere by binding to shugoshin 1 and 2 (Sgo1/Sgo2) (*Kitajima et al., 2006*; *Riedel et al., 2006*; *Rivera et al., 2012*; *Tang et al., 2006*; *Tanno et al., 2010*; *Xu et al., 2009*). The crystal structure of Sgo1 bound to PP2A-B56 has been solved to reveal a bipartite interaction between Sgo1 and the regulatory and catalytic subunits of the PP2A-B56 complex (*Xu et al., 2009*). This interaction is thought to allow centromere-localised PP2A-B56 to counteract various kinases, such as Aurora B, which remove cohesin rings from chromosome arms during early mitosis in higher eukaryotes (*Marston, 2015*). The result is that cohesin is specifically preserved at the centromere where it is needed to resist the

pulling forces exerted by microtubules. As well as preserving cohesion at the centromere, PP2A-B56 is also thought to balance the net level of Aurora B activation in this region (*Meppelink et al., 2015*).

In human cells, B56 isoforms are encoded by five separate genes (B56α, β, γ, δ and ε). The interaction interfaces involved in BubR1 and Sgo1 binding are extremely well conserved between all of these B56 isoforms (*Figure 1—figure supplement 1*). This explains why BubR1 and Sgo1 appear to display no specificity for individual B56 isoforms (*Kitajima et al., 2006*; *Kruse et al., 2013*; *Riedel et al., 2006*; *Xu et al., 2013*; *Xu et al., 2009*), and why these isoforms have been proposed to function redundantly at kinetochores during mitosis (*Foley et al., 2011*).

However, one crucial observation throws doubt over this issue of redundancy: individual B56 isoforms localise differentially to either the kinetochore or centromere in human cells (*Meppelink et al., 2015*; *Nijenhuis et al., 2014*). It is therefore not easy to reconcile this differential localisation with the evidence presented above, which implies that the centromere and kinetochore receptors for B56 do not display any selectivity for individual isoforms. This caused us to readdress the question of redundancy and isoform specificity in human cells.

## Results

### PP2A-B56 isoforms have specific roles at the centromere and kinetochore during mitosis

PP2A-B56 isoform localisation to the centromere and kinetochore was visualised in nocodazole-arrested HeLa Flp-in cells expressing YFP-tagged B56 subunits. This revealed that while some B56 isoforms localise predominantly to the sister kinetochore pairs marked by Cenp-C (B56γ and δ), others localise mainly to the centromere defined by Sgo2 (B56α and ε), and one isoform displayed a mixed localisation pattern (B56β) (*Figure 1a,b*). B56 isoforms have been proposed to act redundantly at the kinetochore in human cells (*Foley et al., 2011*), therefore we readdressed this question in light of their differential localisation. B56α and B56γ were chosen as representative members of the centromere and kinetochore-localised pools, respectively, since these isoforms could both be readily detected by western blot analysis of HeLa cell lysates (*Figure 1—figure supplement 2*). Furthermore, both genes were endogenously tagged using CRISPR/Cas9-mediated homologous recombination to demonstrate consistent expression and differential localisation to either the centromere or kinetochore (*Figure 1—figure supplement 3*). All B56 isoforms were then depleted, except for either B56α or B56γ (*Figure 1—figure supplement 2*), to determine whether these endogenous isoforms could support centromere and kinetochore functions.

Centromeric PP2A-B56 is important for maintaining sister chromatid cohesion during mitosis in human cells (*Marston, 2015*). In agreement with our differential localisation data, only the centromere-localised B56α was able to support proper centromeric cohesion (*Figure 1c*). In fact, we observed no difference in the extent of sister chromatid splitting when comparing loss of all B56 isoforms to a situation when only B56γ is retained (*Figure 1c*). Therefore, sister chromatid cohesion can be supported by a B56 isoform that localises primarily to the centromere (B56α), but not by one that localises to the kinetochore (B56γ).

To examine which B56 isoforms can support kinetochore functions, we first focussed on SAC signalling. The SAC is activated at kinetochores by the phosphorylation of 'MELT' repeats on Knl1 by the kinase Mps1 (*London et al., 2012*; *Shepperd et al., 2012*; *Yamagishi et al., 2012*). These phosphorylated repeats recruit a variety of SAC proteins to the kinetochore, which are then assembled into an inhibitory complex that is released into the cytosol to prevent mitotic exit (*Saurin, 2018*). PP2A-B56 antagonises this process, as evidenced by the fact that removal of B56 from kinetochores prevents Knl1-MELT dephosphorylation and delays mitotic exit following Mps1 inhibition in nocodazole (*Espert et al., 2014*; *Nijenhuis et al., 2014*). Therefore, we sought to address whether these effects were dependent on specific B56 isoforms.

As expected, simultaneous depletion of all B56 isoforms enhanced basal Knl1-MELT phosphorylation in nocodazole, delayed MELT dephosphorylation upon Mps1 inhibition with AZ-3146 (*Hewitt et al., 2010*), and prevented mitotic exit under identical conditions (*Figure 1d–f*). Importantly, these effects were all rescued when endogenous B56γ was preserved, but not if only B56α remained (*Figure 1d–f*). Kinetochore PP2A-B56 also has well-established roles in chromosome

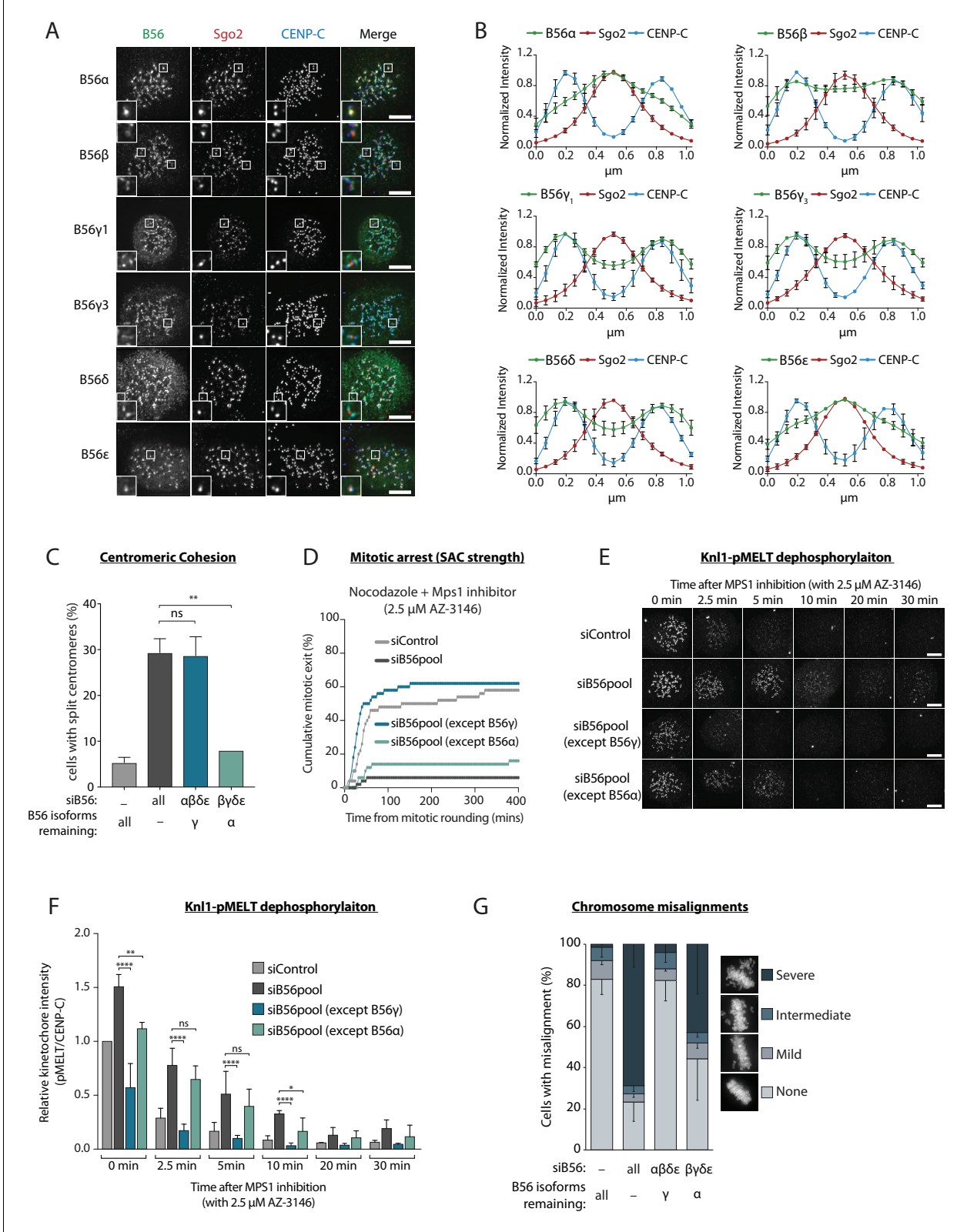

**Figure 1.** A subset of PP2A-B56 complexes control spindle assembly checkpoint silencing and chromosome alignment. (**A** and **B**) Representative images (**A**) and line plots (**B**) of nocodazole-arrested Flp-in HeLa cells expressing YFP-B56 (B56α, B56β, B56γ1, B56γ3, B56δ and B56ε). For line plots, five kinetochore pairs were analysed per cell, for a total of 10 cells per experiment. Graphs represent the mean intensities (±SD) from 3 independent experiments. Intensity is normalized to the maximum signal in each channel in each experiment. (**C–G**) Flp-in HeLa cells treated with siRNA against

*Figure 1 continued on next page*

*Figure 1 continued*

B56pool, all B56 isoforms except B56α, or all B56 isoforms except B56γ, were analysed for sister chromatid cohesion, SAC strength, Knl1-MELT dephosphorylation and chromosomal alignment. (C) Quantification of percentage of chromosome spreads that contain at least one split centromere. Graph represents mean data (+SD) from 3 independent experiments with 50 metaphase spreads quantified per condition per experiment. (D) Time-lapse analysis of cells entering mitosis in the presence of nocodazole and 2.5 μM AZ-3146. The graph represents the cumulative data from 50 cells, which is representative of 3 independent experiments. Representative images (E) and quantification (F) of relative kinetochore intensities of Knl1-pMELT in cells arrested in prometaphase with nocodazole and treated with MG132 for 30 min, followed by 2.5 μM AZ-3146 for the indicated amount of time. 10 cells were quantified per experiment and the graph displays the mean (+SD) of 3 independent experiments. The individual data points for each experiment can be found in the source data. (G) Quantification of chromosome misalignment in cells arrested in metaphase with MG-132. At least 100 cells were scored per condition per experiment and graph represents the mean (-SD) of 3 independent experiments. Misalignments were score as mild (1 to 2 misaligned chromosomes), intermediate (3 to 5 misaligned chromosomes), and severe (>5 misaligned chromosomes). Asterisks indicate significance (*Figure 1c*: Welch's t -test, unpaired, *Figure 1f*: Mann-Whitney test); ns p>0.05, *p≤0.05, **p≤0.01, ****p≤0.0001. Scale bars, 5μm.
DOI: https://doi.org/10.7554/eLife.42619.003
The following figure supplements are available for figure 1:

**Figure supplement 1.** Alignment of B56 isoforms to show that Sgo1 and BubR1 interacting regions are conserved.
DOI: https://doi.org/10.7554/eLife.42619.004
**Figure supplement 2.** Western blot showing knockdown of different B56 isoforms.
DOI: https://doi.org/10.7554/eLife.42619.005
**Figure supplement 3.** Endogenous tagging of B56α and B56γ with YFP confirms centromere and kinetochore localisation.
DOI: https://doi.org/10.7554/eLife.42619.006
**Figure supplement 4.** Overexpression of YFP-B56α can rescue kinetochore functions.
DOI: https://doi.org/10.7554/eLife.42619.007
**Figure supplement 5.** Overexpression of YFP-B56α enhances B56α levels at centromeres and kinetochores.
DOI: https://doi.org/10.7554/eLife.42619.008

alignment where it is needed to antagonise Aurora B and allow initial kinetochore-microtubule attachment to form (*Foley et al., 2011*; *Kruse et al., 2013*; *Suijkerbuijk et al., 2012*; *Xu et al., 2013*). Knockdown of all B56 isoforms produced severe defects in chromosome alignment, as expected, and these effects could be rescued by preserving B56γ, but not B56α (*Figure 1g*). In summary, only the kinetochore-localised B56γ, and not the centromeric B56α, can support SAC silencing and chromosome alignment in human cells.

Overexpression of GFP-B56α has previously been shown to rescue kinetochore-microtubule attachment defects following the depletion of all PP2A-B56 isoforms in human cells (*Foley et al., 2011*). To understand the discrepancy with our data, we performed the same assays as previously, but this time expressing siRNA-resistant YFP-B56α or YFP-B56γ to rescue the knockdown of all endogenous B56 isoforms. Under these conditions, both exogenous B56 isoforms were able to rescue MELT dephosphorylation, SAC silencing and chromosome alignment (*Figure 1—figure supplement 4*). The ability of exogenous YFP-B56α to support kinetochore functions can be explained by the fact that it is highly overexpressed, which leads to elevated centromere and kinetochore levels in comparison to the endogenous YFP-B56α situation (*Figure 1—figure supplement 5*). We therefore conclude B56α acts primarily at the centromere, but it can still function at the kinetochore when overexpressed. In summary, under endogenous conditions, PP2A-B56 isoforms localise differentially to the centromere or kinetochore where they carry out specific roles in sister chromatid cohesion, SAC silencing and chromosomal alignment.

We next sought to determine the molecular explanation for differential B56 isoform localisation. This was difficult to reconcile with existing structural data mapping the interaction between PP2A-B56 and the reported kinetochore and centromere receptors - BubR1 and Sgo1 – since these demonstrate that the key interacting residues are well conserved between all B56 isoforms (*Figure 1—figure supplement 1*) (*Wang et al., 2016a*; *Wang et al., 2016b*; *Xu et al., 2009*). Furthermore, biochemical studies could not detect a difference in association between different B56 isoforms and either BubR1 or Sgo1 (*Kitajima et al., 2006*; *Kruse et al., 2013*; *Xu et al., 2013*). Therefore, we decided to first test whether BubR1 or Sgo1 were the only receptors for B56 at the kinetochore and centromere.

## Sgo2 provides specificity for centromeric B56 recruitment

At the centromere, Sgo1 and Sgo2 can both bind to PP2A-B56 (*Rivera et al., 2012*; *Tanno et al., 2010*; *Xu et al., 2009*), but Sgo1 is considered the primary receptor because it is more important than Sgo2 for protecting cohesion in mitosis (*Huang et al., 2007*; *Kitajima et al., 2005*; *Kitajima et al., 2006*; *Llano et al., 2008*; *McGuinness et al., 2005*; *Rivera et al., 2012*; *Tang et al., 2006*; *Tanno et al., 2010*). However, this critical role in cohesin maintenance could also be explained by PP2A- independent effects (*Hara et al., 2014*). Furthermore, although Sgo1 has been implicated in PP2A-B56 recruitment to centromeres (*Liu et al., 2013a*; *Liu et al., 2013b*; *Nishiyama et al., 2013*; *Tang et al., 2006*), the only study that has directly compared the contribution of Sgo1 and Sgo2 to centromeric PP2A-B56 recruitment, has concluded that Sgo2 is more important (*Kitajima et al., 2006*). We therefore set out to clarify the role of Sgo1 and Sgo2 in controlling the recruitment of B56 isoforms to the centromere in human cells.

Depletion of Sgo2, but not Sgo1, caused a significant reduction in B56α levels at the centromere (*Figure 2a–d*). It is important to note that the quantification in *Figure 2b and d* cannot distinguish between kinetochore and centromere localisation, and whilst Sgo1 depletion did not reduce B56, it did appear to shift its localisation towards the kinetochore (see zoom panel in *Figure 2c*), an effect that has previously been seen by others (*Meppelink et al., 2015*). Line plots analysis, which can quantify localisation across the centromere-kinetochore axis, demonstrates that Sgo1 depletion caused Sgo2 and B56α to spread out from the centromere towards the kinetochore (*Figure 2e*). This is due to inefficient anchoring of Sgo2 at centromeres because combined Sgo1 and Sgo2 depletion completely removed B56α from kinetochores and centromeres (*Figure 2f,g*). We therefore conclude that, as suggested previously by others (*Kitajima et al., 2006*), Sgo2 is the primary centromeric receptor for PP2A-B56 during mitosis. However, Sgo1 also contributes to centromeric B56 localisation primarily by helping to anchor the Sgo2-B56 complex at the centromere, perhaps by bridging an interaction with cohesin rings or by helping to preserve centromeric cohesion (*Hara et al., 2014*; *Liu et al., 2013b*).

We next examined whether specific binding to Sgo1 and/or Sgo2 could explain differential B56 isoform localisation. To address this, we artificially relocalised Sgo1 or Sgo2 to the inner kinetochore, by fusing it to the kinetochore-targeting domain of CENP-B (CB). This location was chosen, even though it partially overlaps with the endogenous centromeric B56 pool, because it is still accessible to Aurora B. This may be important because phosphorylation of Sgo2 by Aurora B has been proposed to be needed for B56 interaction (*Tanno et al., 2010*). Whereas CB-Sgo1 was able to localise additional B56α and B56γ to the inner kinetochore (*Figure 2—figure supplement 1*), CB-Sgo2 was only able to recruit additional B56α (*Figure 2h–k*). To confirm that endogenous Sgo2 displayed selectivity for specific B56 isoforms, we used a Designed Ankyrin Repeat Protein (DARPin) that can bind to GFP with high affinity (*Brauchle et al., 2014*). The DARPin was fused to dCas9 to enable the selective targeting of YFP-tagged B56α or B56γ to a repetitive region on chromosome 7 (Chr7). This assay confirmed that only B56α, and not B56γ, was able to co-recruit endogenous Sgo2 to this region (*Figure 2l,m*). Considering Sgo2 is the primary centromeric receptor for B56 (*Figure 2a,b*) (*Kitajima et al., 2006*), this provides an explanation for why only a subset of B56 isoforms localise to the centromere.

## Sgo1 collaborates with BubR1 to recruit B56 to kinetochores

At the kinetochore, PP2A-B56 binds to a phosphorylated LxxIxE motif in BubR1 (*Kruse et al., 2013*; *Suijkerbuijk et al., 2012*; *Xu et al., 2013*) and this interaction is mediated by a binding pocket on B56 that is completely conserved in all isoforms (*Figure 1—figure supplement 1*) (*Hertz et al., 2016*; *Wang et al., 2016a*; *Wang et al., 2016b*). Therefore, we hypothesised that additional interactions may help to stabilise specific B56 isoforms at the kinetochore. In agreement with this hypothesis, BubR1 depletion or mutation of the LxxIxE binding pocket in B56γ (B56γ$^{H187A}$) reduced but did not completely remove B56γ from kinetochores/centromere (*Figure 3a–d*). This is not due to knockdown efficiency or penetrance of the mutation, because residual B56 could still be detected after BubR1 depletion in B56γ$^{H187A}$ cells (*Figure 3—figure supplement 1a,b*). Interestingly, the remaining B56γ in these situations spreads out between the kinetochore and centromere (*Figure 3e,f*), which implies that B56γ uses additional interactions to be maintained at this region.

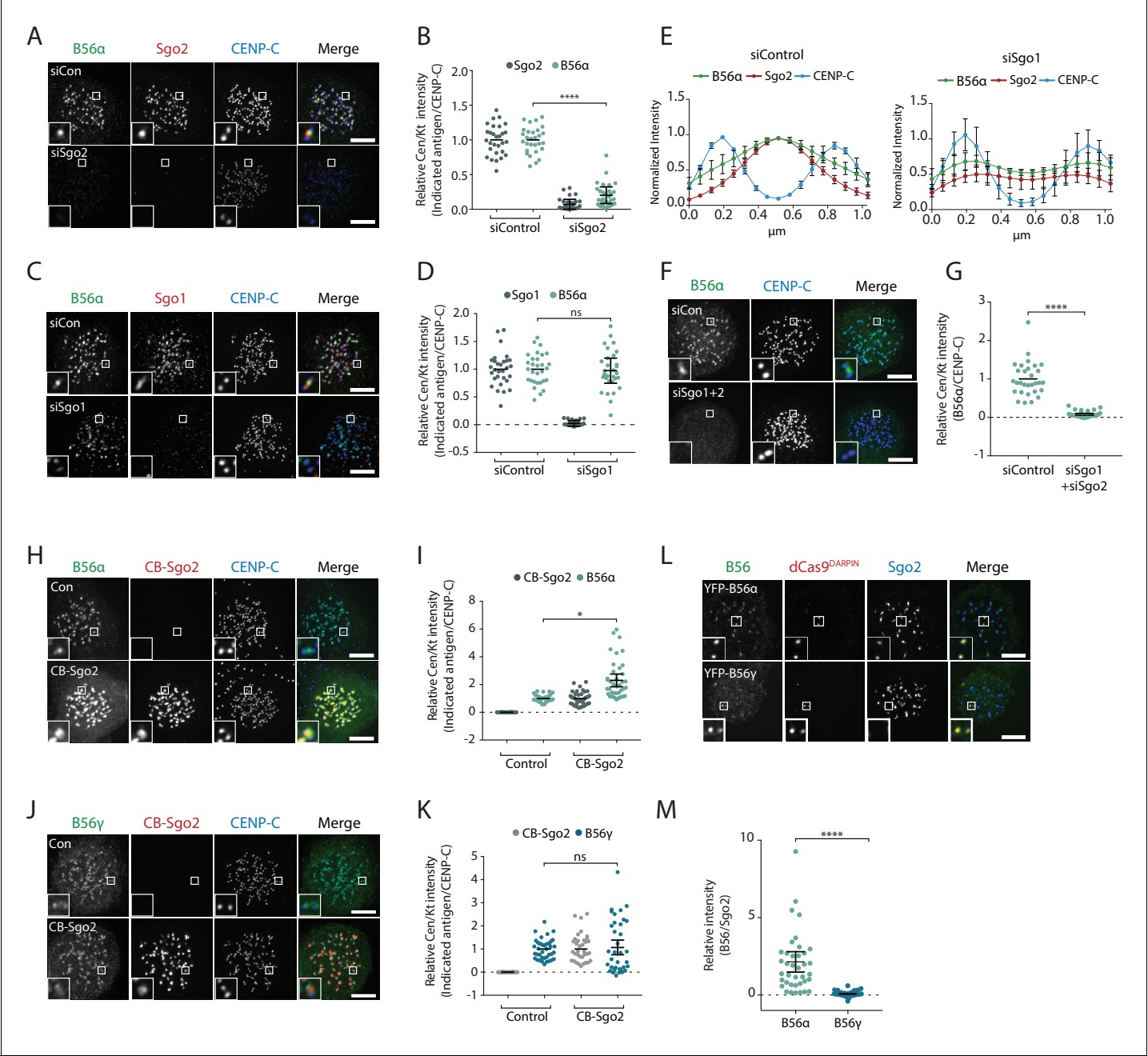

**Figure 2.** Sgo2 specifically localizes B56α to centromeres. (A-G) The effect of Sgo1 and/or Sgo2 knockdown on YFP-B56α localisation in Flp-in HeLa cells. Representative images (A, C, F) and quantifications (B, D, G) of relative kinetochore intensity of B56α in cells arrested in prometaphase with nocodazole after knockdown of Sgo2 (A, B), Sgo1 (C, D), or Sgo1 +Sgo2 (F, G). (E) shows line plots of Sgo2 and B56α localisation following Sgo1 knockdown; 5 kinetochore pairs were analysed per cell, for a total of 10 cells per experiment. Graphs represent the mean intensities (±SD) from 3 independent experiments. Intensity is normalized to the maximum signal present in each channel within the endogenous B56α experiment. (H–M) Flp-in HeLa cells expressing YFP-B56α or YFP-B56γ were transfected with the CB-Sgo2 (H–K) or gChr7 +Cas9 DARPIN (L, M) and analysed for B56 recruitment in cells arrested in prometaphase with nocodazole. (H), (L), and J). are representative images; I) and K) are quantifications of relative centromere/kinetochore intensity of the indicated antigen; and M) is quantification of intensity of Sgo2 over B56 at the Chr7 locus. For all centromere/kinetochore intensity graphs, each dot represents a cell and 10 cells were quantified per experiment for at least 3 independent experiments. The spread of dots indicates the biological variation between individual cells and the errors bars display the variation between the experimental repeats (displayed as -/+SD of the experimental means). Asterisks indicate significance (Mann-Whitney test); ns p>0.05, *p≤0.05, ****p≤0.0001. Scale bars, 5μm.

DOI: https://doi.org/10.7554/eLife.42619.009

*Figure 2 continued on next page*

*Figure 2 continued*

The following figure supplement is available for figure 2:

**Figure supplement 1.** Cenp B-Sgo1 recruits both B56α and B56γ to centromeres.

DOI: https://doi.org/10.7554/eLife.42619.010

A targeted siRNA screen identified critical roles for Knl1 and Bub1, which, when depleted, completely abolished B56γ recruitment to kinetochores (*Figure 3—figure supplement 1c–f*). Knl1 recruits Bub1 to kinetochores, and Bub1 scaffolds the recruitment of BubR1 (*Johnson et al., 2004*; *Overlack et al., 2015*; *Primorac et al., 2013*). However, in addition to this, Bub1 also phosphorylates histone-H2A to localise Sgo1 to histone tails that are adjacent to the kinetochore (*Baron et al., 2016*; *Kawashima et al., 2010*; *Kitajima et al., 2005*; *Liu et al., 2013a*; *Tang et al., 2004*; *Yamagishi et al., 2010*). Since Sgo1 can bind to B56γ (*Figure 2—figure supplement 1*) we examined its role in the kinetochore recruitment of this isoform. Sgo1 depletion reduced B56γ$^{WT}$ at kinetochores and completely removed B56γ$^{H187A}$ (*Figure 3g,h*). Moreover, this was specific for Sgo1, because Sgo2 depletion had no effect (*Figure 3—figure supplement 1g,h*). To test whether this was due to direct binding to Sgo1, we generated a B56γ Sgo1-binding mutant (B56γ$^{ΔSgo1}$), which we confirmed was defective in binding CB-Sgo1 in vivo (*Figure 3—figure supplement 2*). This mutation reduced the recruitment of B56γ$^{WT}$ to kinetochores and completely abolished the recruitment of B56γ$^{H187A}$ (*Figure 3i,j*), in a manner that was similar to the effect of Sgo1 depletion (*Figure 3g,h*). This demonstrates that Bub1 establishes two separate arms that cooperate to recruit B56γ to kinetochores: it binds directly to BubR1, which interacts via its LxxIxE motif with B56γ, and it phosphorylates Histone-H2A to recruit Sgo1, which additionally helps to anchor B56γ at kinetochores.

## B56 isoforms bind differentially to LxxIxE containing motifs during mitosis

The B56-Sgo1 interaction is unlikely to explain B56 isoform specificity at kinetochores, since Sgo1 interacts with both B56α and B56γ when recruited to centromeres (*Figure 2—figure supplement 1*). We therefore focussed on the LxxIxE interaction with BubR1 to quantitatively assess the binding to B56α and B56γ. Immunoprecipitations of equal amounts of B56α and B56γ from nocodazole-arrested cells demonstrated that BubR1 bound preferentially to B56γ (*Figure 4a,b*). Moreover, a panel of antibodies against other LxxIxE containing proteins (*Hertz et al., 2016*), demonstrated that LxxIxE binding was generally reduced in B56α immunoprecipitates (*Figure 4a,b*). B56γ has been shown to display slightly higher affinities for some LxxIxE containing peptides in vitro (*Wu et al., 2017*), which, in principle, could allow this isoform to outcompete B56α for binding. However, a simple competition model is unlikely to explain differential kinetochore localisation, since we observe no change in B56α localisation when all other B56 isoforms are present or knocked down (*Figure 4c, d*). Instead, we favour the hypothesis that binding to LxxIxE motifs is specifically perturbed in PP2A-B56α complexes during prometaphase.

## Residues within a C-terminal loop of B56 determine localisation to the centromere or kinetochore

We next searched for the molecular explanation for differential B56 isoform localisation. To do this, we generated four chimaeras between B56α and B56γ by joining the isoforms in the loops that connect the α-helixes (*Figure 5a*). Immunofluorescence analysis demonstrated that B56γ localisation switched from kinetochores to centromeres in chimaera 4 (*Figure 5b,c*). Furthermore, this region alone (i.e. the region that is different between chimaeras 3 and 4) is sufficient to switch localisation to the centromere when transferred into B56γ, and the corresponding region in B56γ can induce localisation to the kinetochore if transplanted into B56α (*Figure 5—figure supplement 1*). We generated four additional chimaeras to narrow down this region even further to amino acids 405–425 in B56α, which contains an α-helix and a small loop that juxtaposes the catalytic domain in the PP2A-B56γ complex (*Figure 5d–f*) (*Xu et al., 2006*). Importantly, switching just four amino acids within this loop in B56α to the corresponding residues in B56γ (B56α$^{TKHG}$) was sufficient to relocalise B56α from centromeres to kinetochores (*Figure 5g–i*). Furthermore, the B56α$^{TKHG}$ remained functional and holoenzyme assembly was unperturbed (*Figure 5—figure supplement 2*). In summary, a small

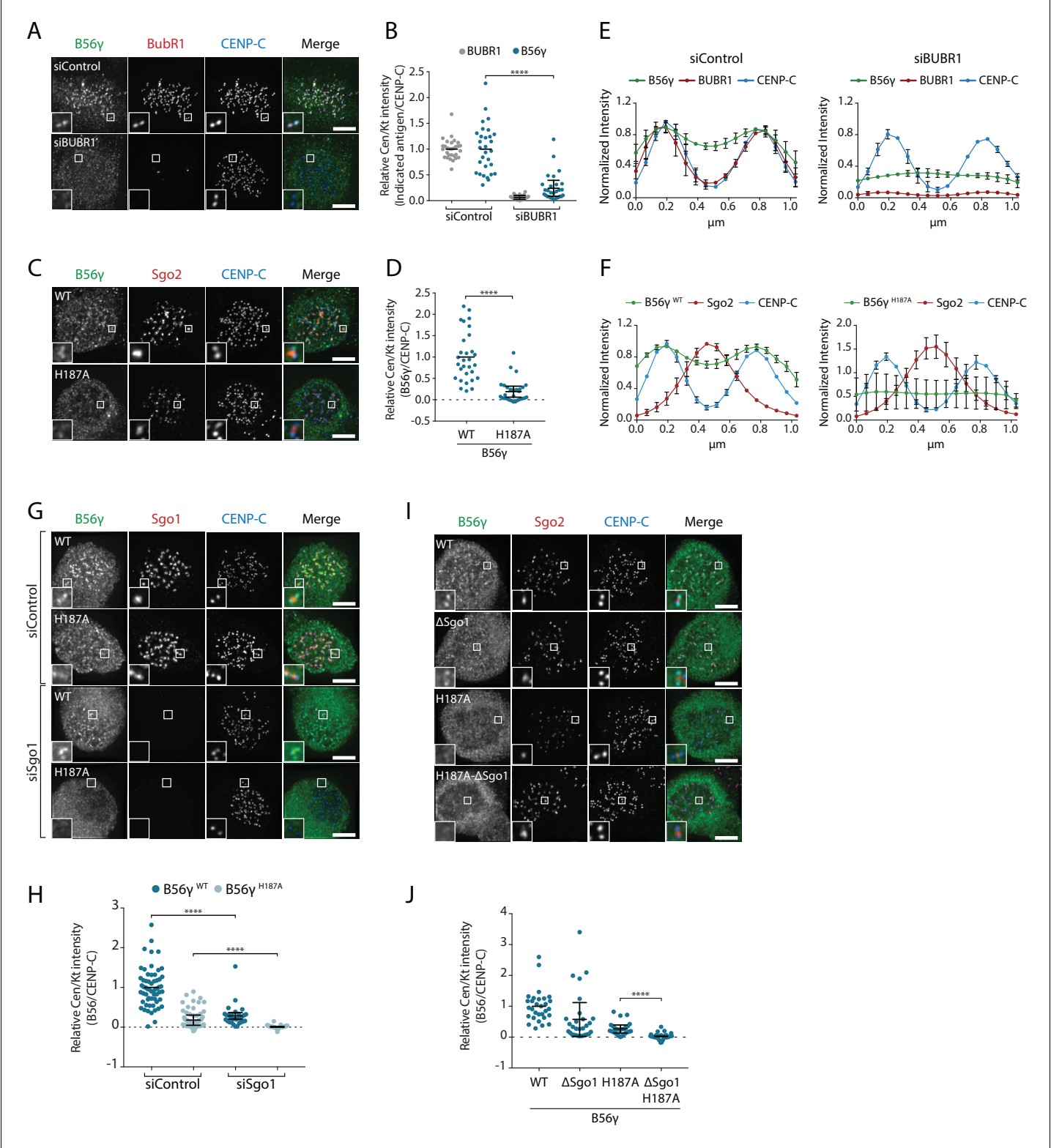

**Figure 3.** BubR1 and Sgo1 localize B56γ to kinetochores. B56γ kinetochore localisation in Flp-in HeLa cells after BubR1 knockdown (**A, B, E**) or mutation of the LxxIxE binding pocket (H187A: **C**), (**D, F**) in cells arrested in prometaphase with nocodazole. For each condition, representative images (**A, C**), quantification of relative centromere/kinetochore levels (**B, D**) and line plot analysis (**E, F**) depicts the levels and distribution of the indicated antigens. (**G–J**): representative images (**G, I**) and quantification of relative centromere/kinetochore intensities (**H, J**) YFP-B56γ WT or H187A following Sgo1 knockdown (**G, H**) or mutation of the Sgo1 binding region (ΔSgo1). For all centromere/kinetochore intensity graphs, each dot represents a cell and 10

*Figure 3 continued on next page*

*Figure 3 continued*

cells were quantified per experiment from at least 3 independent experiments. The spread of dots indicates the biological variation between individual cells and the errors bars display the variation between experimental repeats (displayed as -/+SD of the experimental means). For the line plot analysis, 5 kinetochore pairs were analysed per cell, for a total of 10 cells per experiment. The graph represents the mean intensities (±SD) from at least 3 independent experiments. Intensity is normalized to the maximum signal in each channel in each experiment. Asterisks indicate significance (Mann-Whitney test); ****p≤0.0001. Scale bars, 5µm.

DOI: https://doi.org/10.7554/eLife.42619.011

The following figure supplements are available for figure 3:

**Figure supplement 1.** Knl1 and Bub1 depletion completely removes B56γ from kinetochores.
DOI: https://doi.org/10.7554/eLife.42619.012

**Figure supplement 2.** Mutation of the Sgo1 binding region in B56γ perturbs binding to CB-Sgo1.
DOI: https://doi.org/10.7554/eLife.42619.013

C-terminal loop in B56 defines whether B56 localises to centromeres, via Sgo2, or to kinetochores, via an LxxIxE interaction with BubR1.

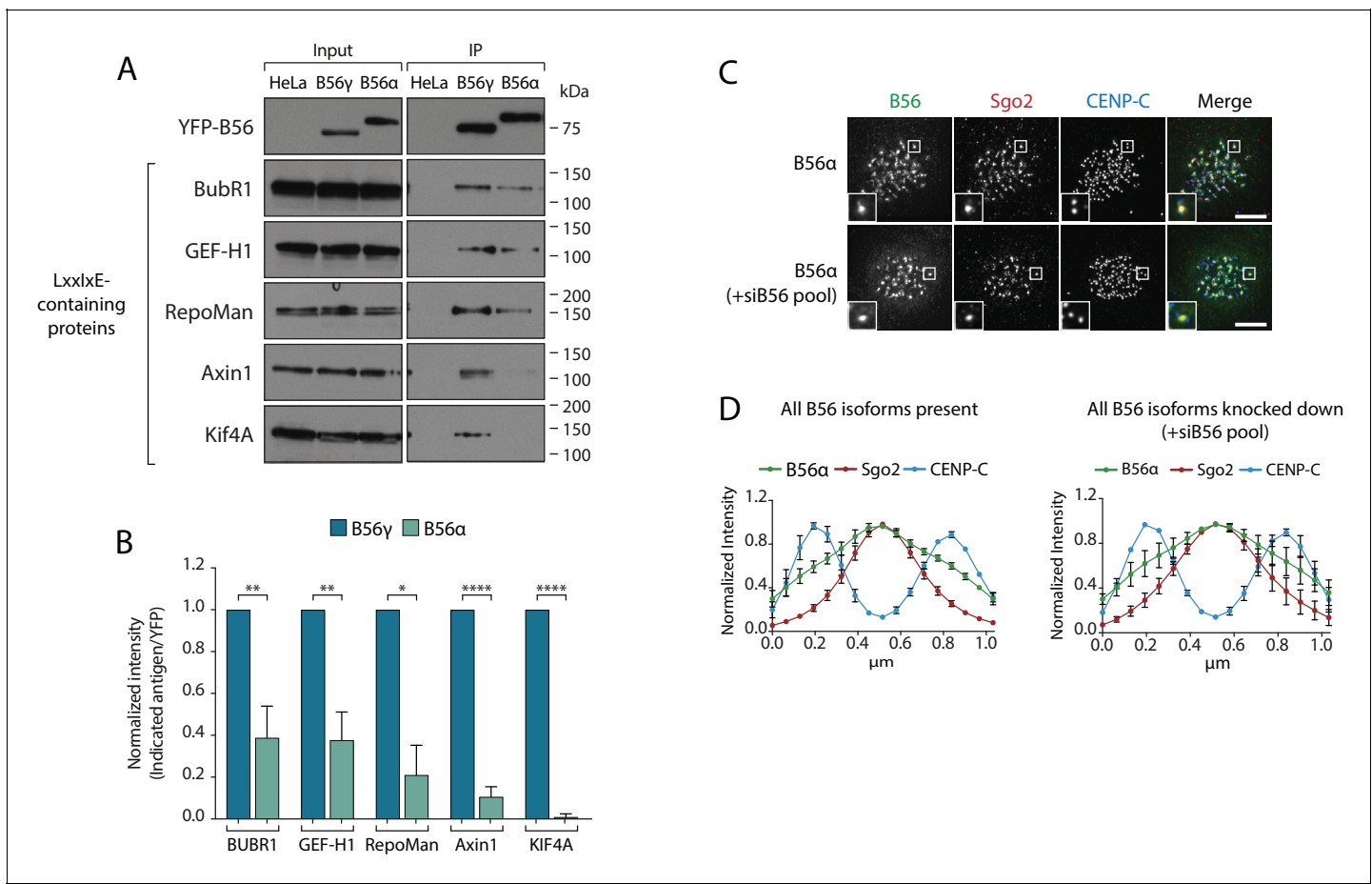

**Figure 4.** Specific binding of B56γ to kinetochores reflects an enhanced ability to bind LxxIxE motifs. (**A**) Immunoblot of the indicated proteins, containing a LxxIxE motif (*Hertz et al., 2016*), following YFP immunoprecipitation from nocodazole-arrested Flp-in HeLa cells expressing YFP-B56α or YFP-B56γ. (**B**) Quantification of the mean normalised intensity (+SD) of the indicated antigens in B56α immunoprecipitates, relative to B56γ immunoprecipitates, from at least 3 independent experiments. Representative images (**C**) and line plot analysis (**D**) of YFP-B56α in Flp-in HeLa cells arrested in nocodazole and treated with the indicated siRNA. Each line plot graph represents the mean intensities (±SD) from 3 independent experiments. 5 kinetochore pairs were analysed per cell, for a total of 10 cells per experiment. Intensity is normalized to the maximum signal in each channel in each experiment. Asterisks indicate significance (Welch's t -test, unpaired); *p≤0.05, **p≤0.01, ****p≤0.0001. Scale bars, 5µm.
DOI: https://doi.org/10.7554/eLife.42619.014

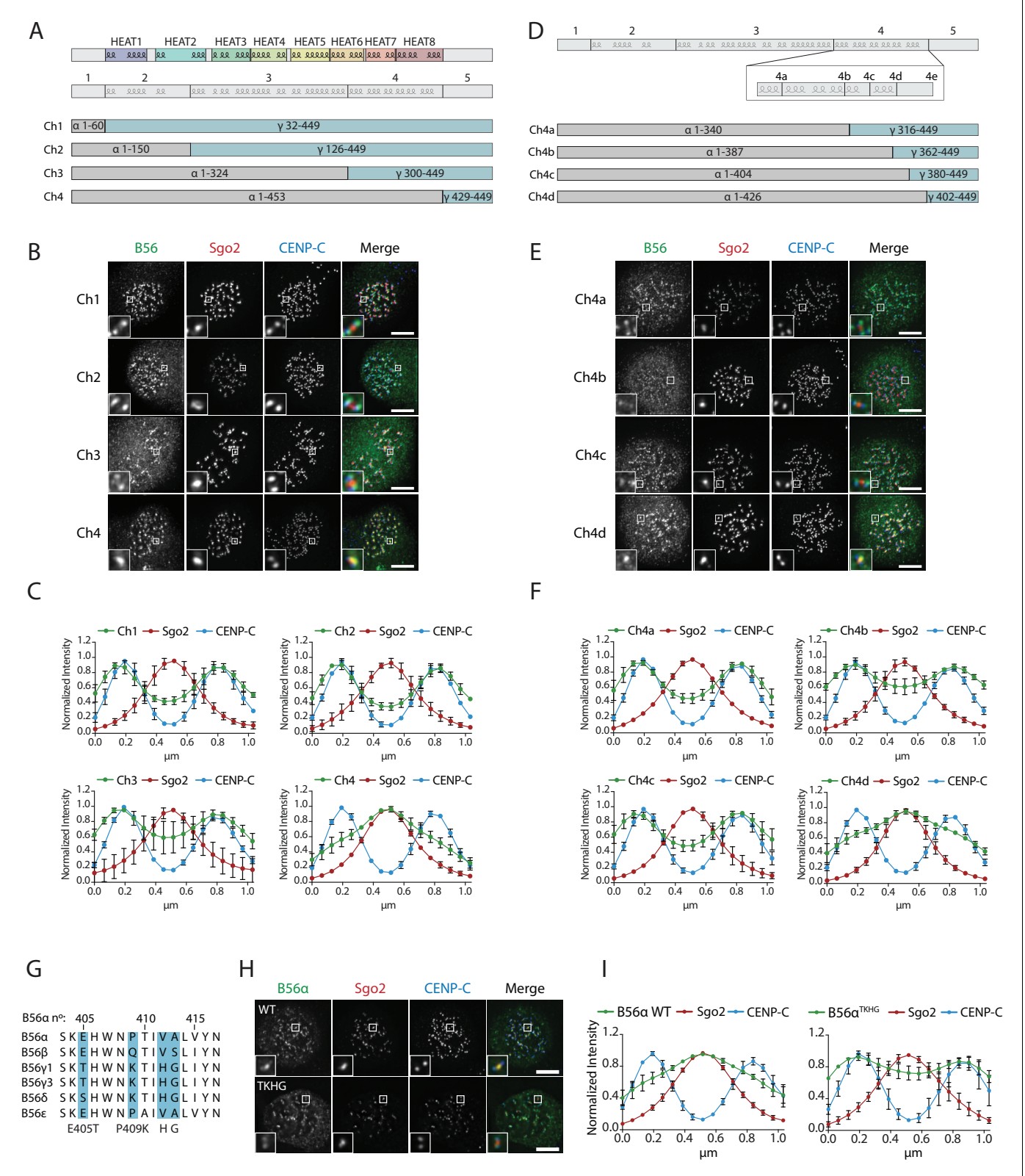

**Figure 5.** A C-terminal loop in B56 specifies B56 localization to centromeres or kinetochores. B56 localisation in B56α-γ chimaeras spanning the entire B56 (Ch1-4: **A–C**), a region at the C-terminus (Ch4a-4d: **D–F**). (**A, D**) Schematic representation of the B56α-γ chimaeras created. Representative images (**B, E**) and line plot analysis (**C, F**) to show the B56 localisation pattern in each chimaera. (**G**). Alignment of B56 isoforms within region 4d that controls centromere/kinetochore localisation. (**G–H**): Effect of 4 point-mutations within region 4d to convert B56α to the correspond B56γ sequence (B56α^TKHG).
*Figure 5 continued on next page*

*Figure 5 continued*

Representative images (H) and line plot analysis (I) of B56α WT or B56α<sup>TKHG</sup> in cells arrested in prometaphase with nocodazole. Each graph represents the mean intensities (±SD) from 3 independent experiments. 5 kinetochore pairs were analysed per cell, for a total of 10 cells per experiment. Intensity is normalized to the maximum signal in each channel in each experiment. Scale bars, 5μm.

DOI: https://doi.org/10.7554/eLife.42619.015

The following figure supplements are available for figure 5:

**Figure supplement 1.** Region four is sufficient to induce B56 localization to the centromere or kinetochore.

DOI: https://doi.org/10.7554/eLife.42619.016

**Figure supplement 2.** Holoenzyme assembly and mitotic exit is unperturbed by B56α TKHG mutation.

DOI: https://doi.org/10.7554/eLife.42619.017

## The C-terminal loop controls Sgo2 binding and LxxIxE motif affinity

We next addressed whether the B56α<sup>TKHG</sup> mutant switched the Sgo2 and LxxIxE binding properties of B56α. In-cell interaction assays demonstrated that B56α<sup>TKHG</sup>, in contrast to B56α<sup>WT</sup>, was not efficiently recruited to the centromere by CB-Sgo2 (*Figure 6a,b*), and was unable to co-recruit endogenous Sgo2 to the repeat region on chromosome 7, when re-localised there using dCas9-DARPin (*Figure 6c,d*). Furthermore, in addition to these effects on Sgo2 interaction, the YFP-B56α<sup>TKHG</sup> mutant showed an enhanced ability to bind LxxIxE containing proteins and, in particular, BubR1, following immunoprecipitation from nocodazole-arrested cells (*Figure 6e,f*). Therefore, we conclude that the small EPVA loop in B56α is necessary for the interaction with Sgo2 and the centromere and, in addition, it is also required to fully repress binding to LxxIxE motifs and the kinetochore. Importantly, this loop is not sufficient to induce either of these effects when transplanted alone into B56γ, because B56γ<sup>EPVA</sup> is not lost from the kinetochore or gained at the centromere (*Figure 6—figure supplement 1a*). Instead, a region immediately C-terminal to the EPVA (amino acids 414–453 in B56α) is also required to induce centromere binding, and a small helix N-terminal to the EPVA (amino acids 374–386 in B56α) is needed to repress kinetochore binding (*Figure 6—figure supplement 1b*). Therefore, although the regions that define centromere and kinetochore localisation overlap at the EPVA loop, they have different distal requirements that demonstrates that they are not identical (*Figure 6g*).

## Discussion

This work demonstrates how different B56 isoforms localise to discrete subcellular compartments to control separate processes during mitosis. Differential B56 isoform localisation has previously been observed in interphase (*McCright et al., 1996*) and during the later stages of mitosis (*Bastos et al., 2014*), which implies that B56 isoforms may have evolved to carry out specific functions, at least in part, by targeting PP2A to distinct subcellular compartments. The differential localisation we observe during prometaphase arises because B56 isoforms display selectivity for specific receptors at the centromere and kinetochore.

The centromeric isoform B56α binds preferentially to Sgo2 via a C-terminal stretch that lies between amino acids 405 and 453 (*Figure 6—figure supplement 1*). A key loop within this region juxtaposes the catalytic domain and contains an important EPVA signature that is critical for Sgo2 binding and is unique to B56α and B56ε. This sequence is also conserved in *Xenopus* B56ε, which has previously been shown to selectively bind to Sgo2, when compared to B56γ (*Rivera et al., 2012*). We therefore propose that a subset of B56 isoforms (B56α and ε) utilize unique motifs to interact with Sgo2 and the centromere during mitosis.

How then can these results be reconciled with the fact that Sgo1 appears to be more important than Sgo2 for the maintenance of cohesion during mitosis (*Huang et al., 2007*; *Kitajima et al., 2005*; *Kitajima et al., 2006*; *Llano et al., 2008*; *McGuinness et al., 2005*; *Rivera et al., 2012*; *Tang et al., 2004*; *Tang et al., 2006*; *Tanno et al., 2010*)? Firstly, it is important to note that Sgo1 can compete with the cohesin release factor, WAPL, for cohesin binding (*Hara et al., 2014*), thereby protecting cohesion independently of PP2A. In addition, Sgo1 could help cells to tolerate the loss of Sgo2, because Sgo2 depletion does not fully remove PP2A-B56 from the centromere, and the pool that remains under these conditions is dependent on Sgo1 (*Figure 2a–g*). Therefore, the residual Sgo1-PP2A-B56α/ε that remains at centromeres following Sgo2 depletion could be sufficient to

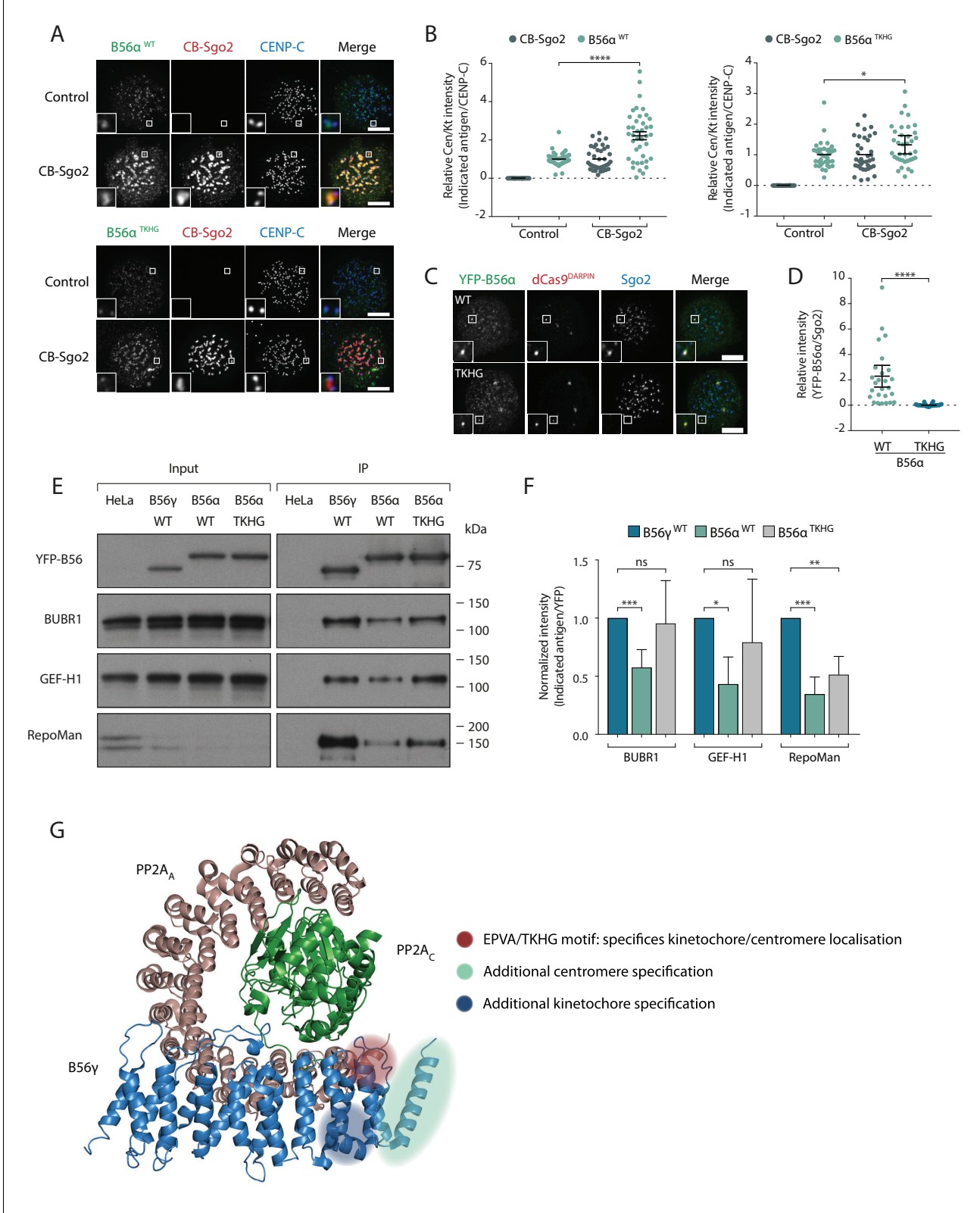

**Figure 6.** A C-terminal loop in B56 regulates binding to Sgo2 and LxxIxE motifs to specify centromere/kinetochore localisation. (**A-D**) Flp-in HeLa cells expressing either YFP-B56α WT or TKHG were transfected with the CB-Sgo2 and analysed for B56 recruitment (**A, B**) or gChr7 +dCas9 DARPIN to assess YFP-B56α:Sgo2 co-localisation (**C, D**). Representative images (**A, C**) and quantification of relative kinetochore intensity (**B**) or intensity of Sgo2 over B56α at the Chr7 locus (**D**). For the intensity graphs in **B**) and **D**), each dot represents a cell and 10 cells were quantified per experiment from at

*Figure 6 continued on next page*

Figure 6 continued

least 3 independent experiments. The spread of dots indicates the biological variation between individual cells and the errors bars display the variation between experimental repeats (displayed as -/+SD of the experimental means). (E) Immunoblot of the indicated antigens following immunoprecipitation of YFP from nocodazole-arrested Flp-in HeLa cells expressing YFP- B56γ, YFP-B56α WT or YFP-B56α-TKHG. (F) Quantification of the mean normalised intensity (+SD) of indicated antigens in B56α WT or B56α TKHG immunoprecipitates, relative to B56γ, from at least 4 experiments. (G) Crystal structure of PP2A-B56γ [accession code 2NPP (*Xu et al., 2006*)] with annotation to indicate the regions that specify localisation to centromeres or kinetochores (see *Figure 6—figure supplement 1* for details of the additional centromere/kinetochore specification regions). Note, the structure is meant only as a guide because the critical regions may be present within B56α, which has not been structurally solved. The B56α WT values are also used in some of the points plotted in *Figure 2m*. Asterisks indicate significance (Mann-Whitney test, except *Figure 6f*: Welch's t -test, unpaired); ns p>0.05, *p≤0.05, **p≤0.01, ***p≤0.001, ****p≤0.0001. Scale bars, 5μm.
DOI: https://doi.org/10.7554/eLife.42619.018

The following figure supplements are available for figure 6:

**Figure supplement 1.** Additional residues from B56α are required to switch the localisation of B56γ.
DOI: https://doi.org/10.7554/eLife.42619.019

**Figure supplement 2.** Sgo2 depletion does not enhance the ability of B56α to bind BubR1 or LxxIxE motifs.
DOI: https://doi.org/10.7554/eLife.42619.020

preserve cohesion. Finally, Sgo1 is needed to preserve Sgo2-PP2A-B56 at the centromere (*Figure 2e*) and it can also bind directly to the SA2–Scc1 complex (*Hara et al., 2014*; *Liu et al., 2013b*; *Tanno et al., 2010*). Therefore, perhaps Sgo1 also helps to position Sgo2-PP2A-B56 so that it can dephosphorylate nearby residues within the cohesin complex. It will be important in future to examine the interplay between Sgo1, Sgo2 and PP2A-B56 at centromeres.

The kinetochore B56 isoforms bind to BubR1 via a canonical LxxIxE motif within the KARD (*Hertz et al., 2016*; *Kruse et al., 2013*; *Suijkerbuijk et al., 2012*; *Xu et al., 2013*). Although the LxxIxE binding pocket is completely conserved in all B56 isoforms (*Figure 1—figure supplement 1*), we observe a striking preference in the binding of B56γ over B56α to many LxxIxE containing proteins during prometaphase (*Figure 4*). We hypothesise that this is due to repressed binding between LxxIxE motifs and B56α during prometaphase, because LxxIxE binding (*Figure 6e,f*) and kinetochore accumulation (*Figure 5h,j*) can both be enhanced by mutation of the EPVA loop in B56α (B56α$^{TKHG}$). We cannot, however, exclude the possibility that the corresponding TKHG sequence in B56γ positively regulates LxxIxE interaction and kinetochore localisation. Considering that this region also controls Sgo2 and centromere binding, a simple explanation could be that Sgo2 interaction obscures the LxxIxE binding pocket. However, this appears unlikely for four reasons: 1) Sgo2 depletion does not relocalise B56α to kinetochores (*Figure 2a,b*), 2) Sgo2 depletion does not enhance the ability of B56α to bind to BubR1 or other LxxIxE motifs during mitosis (*Figure 6—figure supplement 2*), 3) centromere and kinetochore binding can occur together in certain B56α-γ chimaeras (*Figure 6—figure supplement 1b*), and 4) the regions that define each of these localisations do not fully overlap (*Figure 6g*). Although we believe these results imply that Sgo2 is unlikely to block LxxIxE interaction, in vitro experiments with purified components would ultimately be needed to formally rule this out. If not Sgo2, then what could limit the kinetochore accumulation of B56α? We speculate that another interacting partner, or alternatively a tail region within a PP2A-B56 subunit, might obscure or modify the conformation of the LxxIxE binding pocket in PP2A-B56α complexes.

An important additional finding of this work is that Sgo1 contributes to the B56γ signal observed at the kinetochore (*Figure 3g–j*). This likely requires Sgo1 to be bound to histone tails, because it also depends on Bub1, the kinase that phosphorylates histone H2A to recruit Sgo1 (*Baron et al., 2016*; *Kawashima et al., 2010*; *Kitajima et al., 2005*; *Liu et al., 2013a*; *Tang et al., 2004*; *Yamagishi et al., 2010*) (*Figure 3—figure supplement 1*). It is not currently clear whether a Sgo1: PP2A-B56γ complex simply contributes to the signal observed at kinetochores or whether it may help to physically retain BubR1:PP2A-B56 at the kinetochore, for example, by directly interacting with the BubR1:PP2A-B56 complex. The interfaces between BubR1-B56 and Sgo1-PP2A-B56 do not appear to be overlapping, at least based on current structural data (*Wang et al., 2016a*; *Wang et al., 2016b*; *Xu et al., 2009*), which implies that Knl1-bound BubR1-B56 could potentially be anchored towards histone tails by Sgo1. We were unable to detect Sgo1 in YFP-BubR1 immuno-precipitates (results not shown), however, this could simply reflect an interaction that is either transient or unstable away from kinetochores. It will be important in future to clarify exactly how Sgo1

collaborates with BubR1 to control B56 localisation and, in particular, to determine whether Sgo1 can interact with BubR1:PP2A-B56 complexes directly. If such a complex can exist, then this could have important implications for SAC signalling and tension-sensing.

In summary, the work presented here explains how different members of the PP2A-B56 family function during the same stage of mitosis to control different biological processes. This is the first time that such sub-functionalisation has been demonstrated between isoforms of the same B family. It is currently unclear why such specialisation is necessary or at least preferable to a situation whereby all B56 isoforms operate redundantly, as initially suggested (*Foley et al., 2011*). One possibility is that the use of different B56 isoforms allows PP2A catalytic activity to be regulated differently in specific subcellular compartments: for example, by enabling interactions or post-translational modifications that are specific for the B56 subunits. In this respect, protein inhibitors of PP2A-B56 have been shown to function specifically at the centromere (SET (*Chambon et al., 2013*)) and at the kinetochore (BOD1 (*Porter et al., 2013*))); therefore, it would be interesting to test whether these inhibitors display selectivity for certain PP2A-B56 isoforms. Future studies such as this, which build upon the work presented here, may ultimately help to reveal novel ways to modulate the activity of specific PP2A-B56 complexes. The recent development of selective inhibitors of related PP1 regulatory isoforms to combat neurodegenerative diseases (*Das et al., 2015*; *Krzyzosiak et al., 2018*), provides a proof-of-concept that successful targeting of specific serine/threonine phosphatase isoforms is both achievable and therapeutically valuable.

# Materials and methods

## Key resources table

| Reagent type or resource | Designation | Source or reference | Identifiers | Additional information |
|---|---|---|---|---|
| Cell line (*H.sapiens*) | HeLa Flp-in | *Tighe et al. (2008)* | | |
| Recombinant DNA reagent | pcDNA5-YFP-B56 α, β, $\gamma_1$, $\gamma_3$, δ and ε. | This paper | | B56 from pCEP-4xHA-B56 (Addgene 14532–14537) cloned into pcDNA5-LAP-BubR1WT (*Nijenhuis et al., 2014*), Not1-Apa1 sites. |
| Recombinant DNA reagent | pcDNA5-YFP-B56α−(TKHG) | This paper | | Site-directed mutagenesis of pcDNA5-YFP-B56α: E405T, P409K, V412H, A413G |
| Recombinant DNA reagent | pcDNA5-YFP-B56α-(γ4) | This paper | | See *Figure 5—figure supplement 1* |
| Recombinant DNA reagent | pcDNA5-YFP-B56γ-H187A | This paper | | Site-directed mutagenesis of pcDNA5-YFP-B56γ |
| Recombinant DNA reagent | pcDNA5-YFP-B56γ-ΔSgo1 | This paper | | Site-directed mutagenesis of pcDNA5-YFP-B56γ: Y391F, L394S, M398Q. |
| Recombinant DNA reagent | pcDNA5-YFP-B56γ-H187A-ΔSgo1 | This paper | | Site-directed mutagenesis of pcDNA5-YFP-B56γ-H187A: Y391F, L394S, M398Q. |
| Recombinant DNA reagent | pcDNA5-YFP-B56γ-(α4) | This paper | | See *Figure 5—figure supplement 1* |
| Recombinant DNA reagent | pcDNA5-YFP-B56γ-(α4.1) | This paper | | See *Figure 6—figure supplement 1* |
| Recombinant DNA reagent | pcDNA5-YFP-B56γ-(α4.2) | This paper | | See *Figure 6—figure supplement 1* |
| Recombinant DNA reagent | pcDNA5-YFP-B56γ-(α4.3) | This paper | | See *Figure 6—figure supplement 1* |
| Recombinant DNA reagent | pcDNA5-YFP-B56γ−(EPVA) | This paper | | Site-directed mutagenesis of pcDNA5-YFP-B56γ: T631E, K635P, H638V, G639A. |

*Continued on next page*

*Continued*

| Reagent type or resource | Designation | Source or reference | Identifiers | Additional information |
|---|---|---|---|---|
| Recombinant DNA reagent | pcDNA5-YFP-B56-Ch1 | This paper | | See *Figure 5*. |
| Recombinant DNA reagent | pcDNA5-YFP-B56-Ch2 | This paper | | See *Figure 5*. |
| Recombinant DNA reagent | pcDNA5-YFP-B56-Ch3 | This paper | | See *Figure 5*. |
| Recombinant DNA reagent | pcDNA5-YFP-B56-Ch4 | This paper | | See *Figure 5*. |
| Recombinant DNA reagent | pcDNA5-YFP-B56-Ch4a | This paper | | See *Figure 5*. |
| Recombinant DNA reagent | pcDNA5-YFP-B56-Ch4b | This paper | | See *Figure 5*. |
| Recombinant DNA reagent | pcDNA5-YFP-B56-Ch4c | This paper | | See *Figure 5*. |
| Recombinant DNA reagent | pcDNA5-YFP-B56-Ch4d | This paper | | See *Figure 5*. |
| Recombinant DNA reagent | pcDNA5-vsv-CENP-B-Sgo2-mCherry | This paper | | PCR Sgo2 from pDONR-Sgo2 (gift T.J.Yen) into pcDNA5-vsv-CENP-B-Sgo1-mCherry |
| Recombinant DNA reagent | pcDNA5-vsv-CENP-B-Sgo1-mCherry | *Meppelink et al. (2015)* | | |
| Recombinant DNA reagent | pHAGE-TO-dCas9-DARPIN-flag | This paper | | Progenitor plasmid: pHAGE-TO-dCas9-3xmCherry (Addgene 64108). 3xmCherry replaced with synthesised DARPIN-Flag (*Brauchle et al., 2014*). |
| Sequence-based reagent | gRNA targeting a repetetive region on chromosome 7 | *Chen et al. (2016)* | | GCTCTTATGGTGAGAGTGT |
| Sequence-based reagent | B56 Knockin gRNAs | This paper | | B56a: gatgtcgtcgtcgtcgccgccgg. B56g: gtcaacatctagacttcagcggg |
| Sequence-based reagent | siRNAs | *Foley et al. (2011)* | | B56α (PPP2R5A), 5'-UGAAUGAACUGGUUGAGUA-3'; B56β (PPP2R5B), 5'-GAACAAUGAGUAUAUCCUA-3'; B56γ (PPP2R5C), 5'-GGAAGAUGAACCAACGUUA-3'; B56δ (PPP2R5D), 5'-UGACUGAGCCGGUAAUUGU-3'; B56ε (PPP2R5E), 5'-GCACAGCUGGCAUAUUGUA-3'; |
| Sequence-based reagent | siRNAs | *Kitajima et al. (2006)* | | Sgo2, 5'-GCACUACCACUUUGAAUAA-3'; |
| Sequence-based reagent | siRNAs | Dharmacon, J-015475–12 | | Sgo1, 5'-GAUGACAGCUCCAGAAAUU-3'; |
| Sequence-based reagent | siRNAs | *Nijenhuis et al. (2014)* | | BubR1, 5'-AGAUCCUGGCUAACUGUUC-3' |
| Sequence-based reagent | siRNAs | *Vleugel et al. (2013)* | | Knl1, 5'-GCAUGUAUCUCUUAAGGAA-3'; Bub1 5'-GAAUGUAAGCGUUCACGAA-3'; |
| Sequence-based reagent | siRNAs | Dharmacon (D-001830) | | Control (GAPDH), 5'-GUCAACGGAUUUGGUCGUA-3' |
| Antibody | Mouse monoclonal anti-GFP (clone 4E12/8) | Peter Parker, Francis Crick Institute | | 1:1000 |

*Continued on next page*

*Continued*

| Reagent type or resource | Designation | Source or reference | Identifiers | Additional information |
|---|---|---|---|---|
| Antibody | Chicken polyclonal anti-GFP | Abcam | Abcam: ab13970, RRID:AB_300798 | 1:5000 |
| Antibody | Mouse monoclonal anti-Sgo1 (clone 3C11) | Abnova | Abnova: H001516480M01 | 1:1000 |
| Antibody | Rabbit polyclonal anti-Sgo2 | Bethyl | Bethyl: A301-262A, RRID:AB_890650 | 1:1000 |
| Antibody | Mouse monoclonal anti-BubR1 (clone 8G1) | EMD Millipore | EMD Millipore: 05–898, RRID:AB_417374 | 1:1000 |
| Antibody | Mouse monoclonal anti-VSV (clone P5D4) | Sigma | Sigma: V5507, RRID:AB_261877 | 1:1000 |
| Antibody | Rabbit polyclonal anti-Knl1 | Abcam | Abcam: ab70537, RRID:AB_1209410 | 1:1000 |
| Antibody | Rabbit polyclonal anti-Bub1 | Bethyl | Bethy;l: A300-373A, RRID:AB_2065943 | 1:1000 |
| Antibody | Mouse monoclonal anti-FLAG (clone M2) | Sigma | Sigma: F3165, RRID:AB_259529 | 1:10000 |
| Antibody | Guinea Pig polyclonal anti-Cenp-C | MBL | MBL: PD030 | 1:5000 |
| Antibody | Rabbit polyclonal anti-pMELT-Knl1 (phospho-T943 and -T1155) | *Nijenhuis et al. (2014)* | | 1:1000 |
| Antibody | Rabbit polyclonal anti-GFP | Geert Kops, Hubrecht Institute | | 1:5000 |
| Antibody | Mouse monoclonal anti-B56γ (clone A-11) | Santa Cruz Biotechnology | Santa Cruz Biotechnology: sc-374379, RRID:AB_10988028 | 1:1000 |
| Antibody | Mouse monoclonal anti-B56α (clone 23) | BD Biosciences | BD Biosciences: 610615, RRID:AB_397947 | 1:1000 |
| Antibody | Mouse monoclonal anti-B56δ (clone H-11) | Santa Cruz Biotechnology | Santa Cruz Biotechnology: sc-271363, RRID:AB_10611062 | 1:1000 |
| Antibody | Rabbit polyclonal anti-B56ε | Aviva | Aviva: ARP56694-P50 | 1:1000 |
| Antibody | Mouse monoclonal anti-PPP2CA (clone 1D6) | EMD Millipore | EMD Millipore: 05–421, RRID:AB_309726 | 1:5000 |
| Antibody | Rabbit polyclonal anti-PPP2R1A (clone 81G5) | Cell Signaling Technology | Cell Signaling Technology: 2041, RRID:AB_2168121 | 1:1000 |
| Antibody | Rabbit polyclonal anti-BubR1 | Bethyl | Bethyl: A300-386A, RRID:AB_386097 | 1:1000 |
| Antibody | Rabbit polyclonal anti-Axin | Cell Signaling Technology | Cell Signaling Technology: C76H11, RRID:AB_2274550 | 1:1000 |
| Antibody | Rabbit polyclonal anti-GEF-H1 | Abcam | Abcam: ab155785 | 1:1000 |

*Continued on next page*

*Continued*

| Reagent type or resource | Designation | Source or reference | Identifiers | Additional information |
|---|---|---|---|---|
| Antibody | Rabbit polyclonal anti-Kif4A | Bethyl | Bethyl: A301-074A, RRID:AB_2280904 | 1:1000 |
| Antibody | Rabbit polyclonal anti-Repoman | Sigma | Sigma: HPA030049, RRID:AB_10600862 | 1:1000 |
| Antibody | Rabbit polyclonal anti-Actin | Sigma | Sigma: A2066, RRID:AB_476693 | 1:5000 |
| Antibody | Mouse monoclonal anti-α-Tubulin (clone B-5-1-2) | Sigma | Sigma: T5168, RRID:AB_477579 | 1:5000 |
| Antibody | Alexa-fluor488 anti-mouse | ThermoFisher Scientific | Invitrogen: A11029, RRID:AB_138404 | 1:1000 |
| Antibody | Alexa-fluor488 anti-rabbit | ThermoFisher Scientific | Invitrogen: A11034, RRID:AB_2576217 | 1:1000 |
| Antibody | Alexa-fluor488 anti-chicken | ThermoFisher Scientific | Invitrogen: A11039, RRID:AB_142924 | 1:1000 |
| Antibody | Alexa-fluor488 anti-guinea pig | ThermoFisher Scientific | Invitrogen: A11073, RRID:AB_142018 | 1:1000 |
| Antibody | Alexa-fluor568 anti-mouse | ThermoFisher Scientific | Invitrogen: A11031, RRID:AB_144696 | 1:1000 |
| Antibody | Alexa-fluor568 anti-rabbit | ThermoFisher Scientific | Invitrogen: A11036, RRID:AB_10563566 | 1:1000 |
| Antibody | Alexa-fluor647 anti-guinea pig | ThermoFisher Scientific | Invitrogen: A21450, RRID:AB_141882 | 1:1000 |
| Antibody | HRP-anti-mouse | Bio-Rad | Bio-Rad: 170–6516, RRID:AB_11125547 | 1:2000 |
| Antibody | HRP-anti-rabbit | Bio-Rad | Bio-Rad: 170–6515, RRID:AB_11125142 | 1:5000 |
| Chemical compound, drug | AZ-3146 | Selleckchem | Selleckchem: S2731 | |
| Chemical compound, drug | Calyculin A | LC labs | LC labs: C-3987 | |
| Chemical compound, drug | 4,6-diamidino-2-phenylindole (DAPI) | Sigma | Roche: 10236276001 | |
| Chemical compound, drug | Dulbecco's Modified Eagle Medium (DMEM) | ThermoFisher Scientific | Gibco: 41966029 | |
| Chemical compound, drug | Doxycycline hyclate | Sigma | Sigma: D9891 | |
| Chemical compound, drug | Fetal Bovine Serum | ThermoFisher Scientific | Life Technologies: 10270106 | |
| Chemical compound, drug | GFP-Trap magnetic beads | Chromotek | Chromotek: GTMA-20 | |
| Chemical compound, drug | Hygromycin B | Santa Cruz Biotechnology | Santa Cruz Biotechnology: sc-29067 | |
| Chemical compound, drug | Lipofectamine RNAiMax | ThermoFisher Scientific | Invitrogen: 13778150 | |
| Chemical compound, drug | Nocodazole | EMD Millipore | EMD Millipore: 487928 | |
| Chemical compound, drug | MG132 | Selleckcem | Selleckchem: S2619 | |
| Chemical compound, drug | Opti-MEM reduced serum medium | ThermoFisher Scientific | Gibco: 31985–047 | |

*Continued on next page*

*Continued*

| Reagent type or resource | Designation | Source or reference | Identifiers | Additional information |
|---|---|---|---|---|
| Chemical compound, drug | penicillin/streptomycin | ThermoFisher Scientific | Gibco: 15070–063 | |
| Chemical compound, drug | RO-3306 | Tocris | Tocris: 4181 | |
| Chemical compound, drug | Thymidine | Sigma | Sigma: T1895 | |
| Software, algorithm | Kinetochore quantification macro | *Saurin et al. (2011)* | | Software, |
| Algorithm | Multicolor Line plot quantification macro | Kees Straatman (University of Leicester) with modification by Balaji Ramalingam (University of Dundee) | | |
| Software, algorithm | Quantification of immunoblots | Image Studio Lite (LI-COR Biosciences) | | |
| Software, algorithm | Microscopy image processing | Softworx software, GE Healthcare | | |
| Software, algorithm | Microscopy image processing | ImageJ, National Institutes of Health | | |

## Cell culture and reagents

HeLa Flp-in cells (*Tighe et al., 2008*), stably expressing a TetR, were authenticated by STR profiling (Eurofins) and cultured in DMEM supplemented with 9% tetracycline-free FBS, 50 µg/mL penicillin/ streptomycin and 2 mM L-glutamine. All cell lines were routinely screened (every 4–8 weeks) to ensure they were free from mycoplasma contamination. All HeLa Flp-in cells stably expressing a doxycycline-inducible construct were derived from the HeLa Flp-in cell line by transfection with the pCDNA5/FRT/TO vector (Invitrogen) and the FLP recombinase, pOG44 (Invitrogen), and cultured in the same medium but containing 200 µg/mL hygromycin-B. Plasmids were transfected using Fugene HD (Promega) according to manufacturer's protocol. 1 µg/mL doxycycline was added for ≥16 hr to induce protein expression in the inducible cell lines. Thymidine (2 mM) and nocodazole (3.3 µM) were purchased from Millipore, MG132 (10 µM) and AZ-3146 from Selleck Chemicals, doxycycline (1 µg/mL) from Sigma, 4,6- diamidino-2-phenylindole (DAPI, 1:50000) from Invitrogen, calyculin A (10 µM in 10% EtOH) from LC labs, RO-3306 (10 µM) from Tocris and hygromycin-B from Santa Cruz Biotechnology.

## Plasmids and cloning

pCDNA5-YFP -B56α, β, γ1, γ3, δ and ε were amplified from pCEP-4xHA-B56 (Addgene plasmids 14532–14537; deposited by D. Virshup, Duke-NUS Graduate Medical School, Singapore) and subcloned into pCDNA5-LAP-BubR1[WT] (*Nijenhuis et al., 2014*) through Not1 and Apa1 restriction sites. B56γ1 and B56γ3 were corrected to start on M1 and not 11, and the R494L mutation in B56γ3 was corrected. pCDNA5-YFP-B56α and pCDNA5-YFP-B56γ1 were made siRNA-resistant by site-directed mutagenesis (silent mutations in the coding sequence for E102 and L103 in B56α, and T126 and L127 in B56γ). All B56α and B56γ1 mutants were created by site-directed mutagenesis from pCDNA5-YFP-B56α and pCDNA5-YFP-B56γ1, respectively. The B56α–γ chimeras were generated by Gibson assembly with pCDNA5-YFP-B56α and pCDNA5-YFP-B56γ used as templates for the PCR reaction. vsv-CENP-B-Sgo1-mCherry (*Meppelink et al., 2015*) was used to make vsv-CENP-B-Sgo2-mCherry, by removing Sgo1 and adding Sgo2 via Gibson assembly from pDONR-Sgo2 (a gift from T. J. Yen). The Sgo1 binding mutant in B56γ (B56γ[ΔSgo1]) was created by site directed mutagenesis to create three mutations: Y391F, L394S and M398Q. The dCas9-DARPIN-flag was created by digesting pHAGE-TO-dCas9-3xmCherry (Addgene #64108) with BamHI and XhoI to remove 3xmCherry and replace with a synthesised DARPIN-flag that binds to GFP with high affinity (*Brauchle et al., 2014*). The gRNA targeting a repetitive region on chromosome seven was generated by PCR mutagenesis to introduce the gRNA sequence (GCTCTTATGGTGAGAGTGT (*Chen et al., 2016*)) into the pU6 vector.

## Gene knockdowns

Cells were transfected with 20 nM siRNA using Lipofectamine RNAiMAX Transfection Reagent (Life Technologies) according to the manufacturer's instructions. For simultaneous knockdown of all B56 isoforms (B56pool) the single B56 isoform siRNA were mixed at equimolar ratio of 20 nM each. The siRNA sequences used in this study are as follows: B56α (PPP2R5A), 5'-UGAAUGAACUGGUUGAG UA-3'; B56β (PPP2R5B), 5'-GAACAAUGAGUAUAUCCUA-3'; B56γ (PPP2R5C), 5'-GGAAGAUGAAC-CAACGUUA-3'; B56δ (PPP2R5D), 5'-UGACUGAGCCGGUAAUUGU-3'; B56ε (PPP2R5E), 5'-GCA-CAGCUGGCAUAUUGUA-3'; Sgo1, 5'-GAUGACAGCUCCAGAAAUU-3'; Sgo2, 5'-GCACUACCAC UUUGAAUAA-3'; BubR1, 5'-AGAUCCUGGCUAACUGUUC-3'; Knl1, 5'-GCAUGUAUCUCUUAAG-GAA-3'; Bub1 5'-GAAUGUAAGCGUUCACGAA-3'; Control (GAPDH), 5'-GUCAACGGAUUUGGUCG UA-3';. All siRNA oligos were custom made and purchased from Sigma, except for Sgo1, which was ordered from Dharmacon (J-015475–12).

## Expression of B56 isoforms

For reconstitution of B56 isoforms or mutants, HeLa Flp-in cells were transfected with 100 nM B56pool or mock siRNA and, in some experiments, 20 nM additional control, Sgo1, Sgo2, BubR1, Bub1 or Knl1 siRNA. Cells were transfected with the appropriate siRNA for 16 hr, after which they were arrested in S phase for 24 hr by addition of thymidine. Subsequently, cells were released from thymidine for 8–10 hr and arrested in prometaphase by the addition of nocodazole. YFP-B56 expression was induced by the addition of doxycycline during and following the thymidine block. For BubR1 knockdowns and for all chromosome alignment assays, cells were released from thymidine for 6.5 hr and arrested at the G2/M boundary with RO3306 for 2 hr. Cells were then released into nocodazole (BubR1 experiments) or normal growth media (alignment assays) for 15 mins before MG132 was then added for 30 mins to prevent mitotic exit. For alignment assays, this is critical to analyse the synchronous alignment of mitotic cells over a 45 min period.

## Chromosome spreads to analyse centromeric cohesion

Hela-FRT cells were transfected with B56pool, B56βγδε, B56αγδε or control siRNA for 16 hr, treated with thymidine for 24 hr and released into normal growth media for 6.5 hr. Cells were then arrested at the G2/M boundary with RO3306 for 2 hr before release into nocodazole for 1 hr. Mitotic cells were isolated and incubated with hypotonic buffer (20 mM Hepes (pH7.0), 1 mM $MgCl_2$, 20 mM KCl, 2 mM $CaCl_2$) for 10 min at room temperature before being spun onto slides using a Cellspin cytocentrifuge (Tharmac). Slides were airdried for 1 min and then fixed in 4% formaldehyde in PBS for 10 min at room temperature. Blocking and immunofluorescence staining (for Cenp-C to visualise split kinetochore pairs) was carried out as described below. The percentage of cells with at least one split sister kinetochore pair was quantified.

## In-cell protein-protein interaction assay using dCas9 or CB-Sgo1/2

Cells were transfected with dCas9-DARPIN-flag and a guide RNA that targets a repetitive region on chromosome 7 (at 1:3 ratio of dCas9:gRNA). Doxycycline was added to induce YFP-B56 isoform expression and 48 hr later cells arrested in mitosis with nocodazole were fixed, stained and imaged for co-localisation of YFP-B56 isoforms and Sgo2. Only cells containing defined Flag-dCas9 spots that also co-recruited YFP-B56 were imaged. The majority of these spots recruited YFP-B56, but the dCas9 spots themselves were only readily detectable in mitotic cells. For the CB-Sgo1/2 expression experiments, the endogenous Sgo1/2 was still present during these assays.

## CRISPR/Cas9 knock-in

800 base pair homology arms that span left and right of the start codon of B56α and B56γ were custom synthetized by Biomatik. A NaeI (B56γ)/SwaI (B56α) restriction site was place between the homology arms and used to insert a YFP tag by Gibson assembly. Guides were designed to span the start codon (using http://crispr.mit.edu/) so that their complementary sequences are interrupted following successful homologous recombination. Flp-in HeLa Cas9 cells were generated and transfected with the YFP-homology arm vector and guide RNAs (B56α: gatgtcgtcgtcgtcgccgccgg B56γ: gtcaacatctagacttcagcggg) in a 1:1 ratio. Cas9 expression was then induced by addition of

doxycycline and FACS was performed 2 weeks later to sort cells and enrich for the YFP-expressing population.

## Live-cell imaging and immunofluorescence

For time-lapse analysis, cells were plated in 24-well plates, transfected and imaged in a heated chamber (37°C and 5% CO2) using a 10x/0.5 NA on a Zeiss Axiovert 200M Imaging system, controlled by Micro-manager software (open source: https://www.micro-manager.org/). Images were acquired with a Hamamatsu ORCA-ER camera every 4 min using $2 \times 2$ binning. For immunofluorescence, cells were plated on High Precision 1.5H 12 mm coverslips (Marienfeld). Following the appropriate treatment, cells were pre-extracted with 0.1% Triton X-100 in PEM (100 mM Pipes, pH 6.8, 1 mM $MgCl_2$ and 5 mM EGTA) for 1 min followed by addition of 4% PFA/PBS for 2 min; cells were subsequently fixed with 4% paraformaldehyde in PBS for 10 min. Coverslips were washed with PBS and blocked with 3% BSA in PBS + 0.5% Triton X-100 for 30 min, incubated with primary antibodies for 16 hr at 4°C, washed three times with PBS and incubated with secondary antibodies plus DAPI for an additional 2–4 hr at room temperature in the dark. Washed coverslips were then mounted on a glass slide using ProLong antifade reagent (Molecular Probes). All images were acquired on a DeltaVision Core or Elite system equipped with a heated 37°C chamber, with a 100x/1.40 NA U Plan S Apochromat objective using softWoRx software (Applied precision). Images were acquired at $1 \times 1$ binning using a CoolSNAP HQ2 camera (Photometrics) and processed using softWorx software and ImageJ (National Institutes of Health). All images displayed are maximum intensity projections of deconvolved stacks. All displayed immunofluorescence images were chosen to most closely represent the mean quantified data.

## Image quantifications

For kinetochore quantification of immunostainings, all images within an experiment were acquired with identical illumination settings and analysed using ImageJ (for experiments in which ectopic proteins were expressed, cells with comparable levels of exogenous protein were selected for analysis). Kinetochore quantification was performed as previously (*Saurin et al., 2011*). For quantification of B56 localization, The Cenp-C channel was used to choose 5 random kinetochore pairs per cell that lie on the same 0.2 µm Z-plane. A line was then drawn through the kinetochore pairs (using ImageJ), with the first Cenp-C kinetochore peak at 0.2 µm from the start of the line. An ImageJ macro (created by Kees Straatman, University of Leicester and modified by Balaji Ramalingam, University of Dundee) was used to simultaneously measure the intensities in each channel across the line. The signal from the five kinetochore pairs was averaged and normalized to the maximum signal in each channel. For chromosome alignment assays, misalignments were score as mild (1 to 2 misaligned chromosomes), intermediate (3 to 5 misaligned chromosomes), and severe (>5 misaligned chromosomes). For mitotic exit assays, time from entry into mitosis (defined by the rounding up of the cell) to mitotic exit (defined by the separation of the sister chromatids or flattening down of the cell in nocodazole +AZ-3146) were recorded for 50 cells. Data is presented as cumulative percentage of mitotic exit over time.

## Immunoprecipitation and immunoblotting

Flp-in HeLa cells were treated with thymidine and doxycycline for 24 hr and subsequently released into fresh media supplemented with doxycycline and nocodazole for 16 hr. Mitotic cells were isolated by mitotic shake off and lysed in lysis buffer (50 mM Tris, pH 7.5, 150 mM NaCl, 0.5% TX-100, 1 mM $Na_3VO_4$, 5 mM ß-glycerophosphate, 25 mM NaF, 10 nM Calyculin A and complete protease inhibitor containing EDTA (Roche)) on ice. The lysate was incubated with GFP-Trap magnetic beads (from ChromoTek) for 2 hr at 4°C on a rotating wheel in wash buffer (same as lysis Buffer, but without TX-100) at a 3:2 ratio of wash buffer:lysate. The beads were washed 3x with wash buffer and the sample was eluted according to the protocol from ChromoTek. Samples were them processed for SDS-Page and immunoblotting using standard protocols.

## Quantification of immunoblots

For quantification of relative immunoprecipitation levels, scanned immunoblots were analyzed using Image Studio Lite (LI-COR Biosciences). A rectangle of the same size was drawn around each band

and the intensity within the band (minus the background) was calculated. The immunoprecipitated protein was used as a control, and each band was normalized to it.

## Antibodies

All antibodies were diluted in 3% BSA in PBS. The following primary antibodies were used for immunofluorescence imaging (at the final concentration indicated): mouse α-GFP (clone 4E12/8, a gift from P. Parker; 1:1000), chicken α-GFP (ab13970, Abcam; 1:5000), mouse α-Sgo1 (clone 3C11, H00151648-M01, Abnova; 1:1000), rabbit α-Sgo2 (A301-262A, Bethyl; 1:1000), mouse α-BubR1 (clone 8G1, 05–898, Upstate/Millipore; 1:1000), mouse α-VSV (clone P5D4, V5507, Sigma; 1:1000), rabbit α-Knl1 (ab70537, Abcam; 1:1000), rabbit α-Bub1 (A300-373A, Bethyl; 1:1000), mouse α-FLAG (clone M2, F3165, Sigma, 1:10000) guinea pig α-Cenp-C (PD030, MBL; 1:5000) and rabbit α-pMELT-Knl1 directed against T943 and T1155 of human Knl1 (*Nijenhuis et al., 2014*), 1:1000). Secondary antibodies used were highly-cross absorbed goat α-rabbit, α-mouse, α-guinea pig or α-chicken coupled to Alexa Fluor 488, Alexa Fluor 568, or Alexa Fluor 647 (Life Technologies); all were used at 1:1000.

The following antibodies were used for western blotting (at the final concentration indicated): rabbit α-GFP (custom polyclonal, a gift from G. Kops; 1:5000), mouse α-B56γ (clone A-11, sc-374379, Santa Cruz Biotechnology; 1:1000), mouse α-B56α (clone 23, 610615, BD; 1:1000), mouse α-B56δ (clone H-11, sc-271363, Santa Cruz, 1:1000), rabbit α-B56ε (ARP56694-P050, Aviva, 1:1000), mouse α-PPP2CA (clone 1D6, 05–421, Millipore; 1:5000) and rabbit α-PPP2R1A (clone 81G5, #2041, CST; 1:1000), rabbit α-BubR1 (A300-386A, Bethyl; 1:1000), rabbit α-Axin (C76H11, CST; 1:1000), rabbit α-GEF-H1 (155785, Abcam; 1:1000), rabbit α-Kif4a (A301-074A, Bethyl; 1:1000), rabbit α-RepoMan (HPA030049, Sigma; 1:1000) and rabbit α-Actin (A2066, Sigma; 1:5000) and mouse α-alpha-Tubulin (clone B-5-1-2, T5168, Sigma, 1:5000). Secondary antibodies used were goat α-mouse IgG HRP conjugate (Bio-Rad; 1:2000) and goat α-rabbit IgG HRP conjugate (Bio-Rad; 1:5000).

## Statistical tests

Mann-Whitney U test was performed to compare experimental groups in all kinetochore/centromere quantification graphs, whereas two-tailed, unpaired t-test with Welch's correction was performed to compare experimental groups in all other graphs (using Prism seven software). The n numbers for kinetochore/centromere quantification statistics were derived from the individual cells (i.e. biological replicates), which were always from at least three separate experiments (i.e. technical replicates) with similar results. The n numbers for the statistics in all other graphs were defined by the number of experimental repeats. The SD bars displayed in each graph shows the variation between the means of the experimental repeats. The statistical comparisons most pertinent for the conclusions are shown in the figures and legends. The original data for all experiments displayed in graphs can be found in the raw data source file, which also contains the actual statistical values.

## Acknowledgements

This work was funded by Cancer Research UK (C47320/A21229 to ATS) and Tenovus Scotland. We thank staff at the Dundee Imaging, Sequencing, Flow Cytometry and Cell Sorting facilities. We also thank Stephen Taylor for providing the HeLa Flp-in cell line, Timothy Yen and Susanne Lens for providing plasmids, Geert Kops for antibodies, and Iain Cheeseman for helpful discussions.

## Additional information

### Funding

| Funder | Grant reference number | Author |
| --- | --- | --- |
| Tenovus Scotland | T14-19 | Giulia Vallardi |
| Cancer Research UK | C47320/A21229 | Lindsey A Allan<br>Adrian T Saurin |

The funders had no role in study design, data collection and interpretation, or the decision to submit the work for publication.

## Author contributions
Giulia Vallardi, Conceptualization, Data curation, Formal analysis, Investigation, Visualization, Writing—review and editing; Lindsey A Allan, Investigation; Lisa Crozier, Methodology; Adrian T Saurin, Conceptualization, Supervision, Funding acquisition, Investigation, Methodology, Writing—original draft, Writing—review and editing

## Author ORCIDs
Adrian T Saurin (iD) http://orcid.org/0000-0001-9317-2255

## Decision letter and Author response
Decision letter https://doi.org/10.7554/eLife.42619.025
Author response https://doi.org/10.7554/eLife.42619.026

## Additional files

### Supplementary files
• Source data 1. The raw data and statistical values from all the individual experiments that are expressed in graphical format. This files contains the raw data and statistical values from all the graphs displayed in *Figures 1–6*; *Figure 1—figure supplements 2*, *4* and *5*; *Figure 2—figure supplement 1*; *Figure 3—figure supplement 2*; *Figure 5—figure supplements 1* and *2*; *Figure 6—figure supplement 1*.
DOI: https://doi.org/10.7554/eLife.42619.021
• Transparent reporting form
DOI: https://doi.org/10.7554/eLife.42619.022

### Data availability
All data generated or analysed during this study are included in the manuscript and supporting files. Source data files have been provided for all figures that contain graphical information, which is every figure except Figure 1—figure supplements 1 and 2, and Figure 6 - figure supplement 2

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
