## [Decision Letter]

Thank you for submitting your article "Division of labour between PP2A-B56 isoforms at the centromere and kinetochore" for consideration by *eLife*. Your article has been reviewed by Andrea Musacchio as the Senior Editor and Reviewing Editor, and three reviewers. The reviewers have opted to remain anonymous.

The reviewers have discussed the reviews with one another and the Reviewing Editor has drafted this decision to help you prepare a revised submission.

Summary:

PP2A occurs mostly as a trimeric enzyme, consisting of a catalytic C-subunit, a scaffolding A-subunit and a regulatory B-subunit. The literature describes four types of structurally unrelated B subunits and their diversity is further augmented by the expression of isoforms. Five different genes encode for the B' (B56) isoform of PP2A. They are highly similar in amino acid sequence suggesting that they share redundant functions. Yet, recent studies in human cells revealed that individual B56 isoforms display distinct localization patterns, i.e. they localize to either the kinetochore or the centromere. In the submitted manuscript, Vallardi et al., investigate the molecular mechanisms underlying this differential localisation and function. They identify a domain in B56 that is responsible for this and show that it modulates the interaction with proteins containing a short linear motif (SLiM). This is probably the most novel observation reported here. Other observations, for instance the role of Sgo2 as the main receptor for centromeric PP2A-B56, are known from previous work (Kitajima et al., 2006), but the present study is systematic and provides a more comprehensive picture. Collectively, the work was praised for being systematic and because it sheds light on the molecular basis of a poorly understood phenomenon. While the analysis is generally strong, a major concern and several other points were discussed.

Essential revisions:

A point emerged strongly in the post-review discussion, namely that quantitative evidence corroborating claims on the role of the newly identified sequence motifs would not simply be nice to have but would strongly reinforce the paper. An analysis in vitro with purified B56a and B56g and peptides containing the relevant sequence motifs should aim to assess whether Sgo2 reduces the binding of B56a to BubR1, whether Sgo1 binds directly to Bubr1-B56, and whether these interactions are affected by mutation of the EPVA or TKHG motifs. Based on a significant body of previous work, we believe that these experiments are feasible and can be carried out in a time compatible with an *eLife* revision.

Other important points:

For some intensity plots, the authors indicate if the observed differences are significant or not (Figure 2, Figure 3B and D, Figure 2—figure supplement 1, Figure 3—figure supplement 2, Figure 6) but for others they do not (Figure 3H and J, Figure 3—figure supplement 1). Please add this information to all the plots.

In some of the presented images, a clear-cut centromeric or kinetochore localization of B5a and B56g, respectively, as compared to the marker proteins Sgo2 and CENP-C, was not immediately recognisable (see for example Figure 1B). This is particularly true for endogenously tagged B56g, which is almost equally abundant at kinetochores and centromeres (see Figure 1—figure supplement 3D). At the very least, the partially overlapping localisation should be acknowledged.

Kinetochore associated B56g, but not centromeric B56a, was sufficient to maintain the kinetochore functions of PP2A-B56 after the knockdown of all other B56 isoforms. The authors should explicitly state here that this cannot be explained by a lower expression level of B56a in these cells, as is only shown later in Figure 4A (but never commented upon). The reverse experiment, addressing whether B56a alone, but not B56g, can maintain the centromeric functions of PP2A-B56 is lacking and the authors could elect to carry it out.

The relative kinetochore intensity (e.g. Figure 1—figure supplement 5C) apparently refers to the combined kinetochore + centromere targeting. The normalised kinetochore intensity then shows the distribution between kinetochores and centromeres (e.g. Figure 1—figure supplement 5D). The y-axes should be re-defined to make this difference clear.

Figure 1—figure supplement 2. Please include the WB for the other B56 isoforms.

In Figure 2 and Figure 2—figure supplement 1Figure, the authors aim to demonstrate that specific binding of B56a to Sgo2 contributes to its centromeric localization. With these figures, the following problems were identified:

Figure 2C: Upon Sgo1 depletion, B56a seems to change its localization from a primarily centromeric localization to a primarily kt localization. Yet, this is not reflected in the quantifications shown in Figure 2D and 2E. Furthermore, Figure 2D gives the impression that Sgo1 is very efficiently depleted in Sgo1RNAi cells. The IF image shown in C gives a different impression. Could the authors please comment on this?

Subsection “Sgo2 provides specificity for centromeric B56 recruitment”: "…, CB-Sgo2 was only able to recruit B56a (Figure 2h-k)." This statement is puzzling. Figure 2J shows hardly any YFP-B56g localization. Why is the localization so different compared to Figure 1 and how was the quantification based on such IF images performed? Importantly, it should be clarified whether this figure and the related Figure 2—figure supplement 1 are carried out under conditions of depletion of Sgo1 and/or Sgo2, as this does not seem to be stated anywhere.

Subsection “Sgo2 provides specificity for centromeric B56 recruitment”: "…CB-Sgo1 was able to localize both B56a and B56g to the inner kinetochore (Figure 2—figure supplement 1)." This statement is not supported by the data. Under wt conditions, B56g already colocalizes with CENP-C as shown in Figure 1A and 1B. Why are the quantifications (Figure 2K and Figure 2—figure supplement 1D) so different for control conditions?

And given that B56g colocalizes with CENP-C under wt conditions, how do the authors come to the conclusion that CB-Sgo1 recruits B56g to inner kts?

5) Along the same line, B56a shows a stronger co-localization with CENP-C under control conditions than B56g (compare Figure 2—figure supplement 1B to Figure 2—figure supplement 1D).

The figure legend of Figure 2—figure supplement 1 (CB-Sgo1 recruits both B56a and B56g to centromeres) may be wrong. Apparently, what is meant is.….to inner Kts.

Figure 3—figure supplement 2: There is hardly any signal for wt B56g (Figure 3—figure supplement 2A) that colocalizes with CENP-C. Could the authors please comment on this?

One of the reviewers raised a concern that the data are presented in a very compact manner that may not be readily accessible to a broad readership. While no specific suggestion was provided, the authors should consider this point and consider changes that will improve readability and raise the paper's general interest.

---

## [Author Response]

*A point emerged strongly in the post-review discussion, namely that quantitative evidence corroborating claims on the role of the newly identified sequence motifs would not simply be nice to have but would strongly reinforce the paper. An analysis* in vitro *with purified B56a and B56g and peptides containing the relevant sequence motifs should aim to assess whether Sgo2 reduces the binding of B56a to BubR1, whether Sgo1 binds directly to Bubr1-B56, and whether these interactions are affected by mutation of the EPVA or TKHG motifs. Based on a significant body of previous work, we believe that these experiments are feasible and can be carried out in a time compatible with an eLife revision.*

We thank the reviewers and editors for their very careful review and valuable comments/suggestions. In general, we agreed that all of the proposed changes would add value to the manuscript. However, as discussed in a subsequent email exchange with the editor, we also believe that the suggestion to use purified components could not be achieved in a timeframe that is compatible with an *eLife* review. This decision was reached for the following reasons:

The proposed work would require the purification of 7 different components (BubR1, B56α, B56γ, PPP2CA, PPP2R1A, Sgo1 and Sgo2) followed by the phosphorylation of one of these components (BubR1) with two kinases (Plk1/Cdk1) in an attempt to generate two stable complexes: (1) Sgo1:pBubR1:PP2A-B56γ, and (2) Sgo2:PP2A-B56α. This second complex could then be tested for its ability to bind to phospho-BubR1. These in vitro assemblies would also then need to be carried out in the presence of peptides against our identified motif to see if complex assembly was prevented. In addition to the obvious time constraints, we were concerned that assembling phospho-dependent interactions in the presence of a purified phosphatase may be difficult, and we would also have no indication as to whether the peptides could adopt a conformation that binds to Sgo1 or Sgo2 with sufficient strength to prevent the key interactions.

Nevertheless, although these in vitro experiments did not seem feasible to us, we did still clearly see the value of trying to address the main questions set out in the review. Therefore, as discussed in our previous email exchange, we have attempted to address these key questions (paraphrased below for clarity) by alternative means:

1) Does Sgo2 inhibit the ability of B56α to bind to BubR1?

2) Does Sgo1 bind directly to the BubR1-B56 complex?

3) Are any effects identified in 1 or 2 dependent on the EPVA/TKHG motif?

4) Can centromeric cohesion be maintained by only B56α, and not B56γ, as our localisation data would predict.

Below is a summary of the new data that is now included in the manuscript to address these key questions:

1) Does Sgo2 inhibit the ability of B56a to bind to BubR1?

We have now examined the effect of Sgo2 depletion on the ability of B56α to bind to BubR1 and other LxxIxE-containing proteins. This new data (now included as Figure 6—figure supplement 2) shows that efficient Sgo2 depletion does not enhance the ability of B56α to bind to either BubR1 or other proteins with an LxxIxE motif. Furthermore, we are unable to detect clear Sgo2 signal in any B56 IPs, even though LxxIxE interactions are clearly different between B56α and B56γ. We now have four independent pieces of data to suggest that Sgo2 does not inhibit B56α binding to BubR1. We copy a paragraph from the new Discussion section that expands on this point:

“…a simple explanation could be that Sgo2 interaction obscures the LxxIxE binding pocket. However, this appears unlikely for four reasons: (1) Sgo2 depletion does not relocalise B56α to kinetochores (Figure 2A,B), (2) Sgo2 depletion does not enhance the ability of B56α to bind to BubR1 or other LxxIxE motifs during mitosis (Figure 6—figure supplement 2), (3) centromere and kinetochore binding can occur together in certain B56α-γ chimaeras (Figure 6—figure supplement 1B), and (4) the regions that define each of these localisations do not fully overlap (Figure 6G). Although we believe these results suggest that Sgo2 is unlikely to block LxxIxE interaction, in vitro experiments with purified components would ultimately be needed to formally rule this out.”

2) Does Sgo1 bind directly to the BubR1-B56 complex?

Current structural data suggests that Sgo1 and B56 could interact simultaneously with BubR1, and our data indicates that a BubR1-B56-Sgo1 complex could potentially exist at kinetochores. Considering that Sgo1 binds directly to the PP2A-B56 complex, we reasoned that we could address this question by immunoprecipitating BubR1-wt or -dKARD (which cannot bind B56). If Sgo1 is detected in BubR1-wt, but not BubR1-dKARD IPs, then this would suggest that BubR1 requires B56 for Sgo1 interaction; thus, implying the existence of a BubR1-PP2A-B56:Sgo1 complex.

We therefore immunoprecipitated BubR1-wt or dKARD from nocodazole-arrested cells and probed for Sgo1 and B56. The figure below shows that we could detect B56γ, but not B56α, in immunoprecipates from only BubR1-wt cells, as predicted. Unfortunately, however, we were unable to detect Sgo1 in any of these IPs, even after scaling up the input and performing rapid purifications (with a GFP-nanobody) to try to detect transient interactions. Therefore, we cannot currently state whether Sgo1 does not form a complex with BubR1-B56 in cells, or whether this complex is simply not stable enough to be detected following immunoprecipitation. We suggest to keep this as “results not shown” and have added the following new paragraph to the discussion:

“We were unable to detect Sgo1 in YFP-BubR1 immunoprecipitates (results not shown), however, this could simply reflect an interaction that is either transient or unstable away from kinetochores. It will be important in future to clarify exactly how Sgo1 collaborates with BubR1 to control B56 localisation and, in particular, to determine whether Sgo1 can interact with BubR1:PP2A-B56 complexes directly.”

We would like to stress that we do not see this question as critical to our study at all. It follows on from a novel observation that we made regarding kinetochore B56 recruitment, but this does not impact on isoform-specific localisation at all (Sgo1 binds all B56 isoforms equally well: see Figure 2—figure supplement 1). That is why we believe this question is suitable to be left as a future direction.

3) Are any effects identified in 1 or 2 dependent on the EPVA/TKHG motif?

As outlined in point 1 above, Sgo2 is very unlikely to affect LxxIxE/BubR1 binding. Therefore, although the EPVA motif is definitely required for Sgo2 binding (Figure 6A-D), this is unlikely to impact on LxxIxE binding. In addition, we believe that the EPVA/TKHG motif cannot strongly influence Sgo1 binding because Sgo1 can bind to both B56α and B56g (Figure 2—figure supplement 1).

4) Can centromeric cohesion be maintained by only B56α, and not B56γ, as our localisation data predicts.

We were actually very surprised that this was not considered an essential revision, given than it is not possible to formally demonstrate a true division of labour between B56 isoforms – as our title states – without this information. Furthermore, in our opinion, this was perhaps the most novel aspect of our study given that there are no previous demonstration that B56 isoforms can control separate cellular processes. This particular advance was not commented on in the summary by the reviewers, which we suspect is the main reason that this point was not seen as essential.

Nevertheless, we now include this important new data in Figure 1c, which demonstrates that, as predicted previously, depletion of all B56 isoforms leads to premature sister chromatid splitting, and this can be fully rescued if B56α is retained, but not if only B56γ remains.

Other important points:

For some intensity plots, the authors indicate if the observed differences are significant or not (Figure 2, Figure 3B and D, Figure 2—figure supplement 1, Figure 3—figure supplement 2, Figure 6) but for others they do not (Figure 3H and J, Figure 3—figure supplement 1). Please add this information to all the plots.

Thank you for pointing out this omission. The correct stats have now been added to all the Figures.

In some of the presented images, a clear-cut centromeric or kinetochore localization of B5a and B56g, respectively, as compared to the marker proteins Sgo2 and CENP-C, was not immediately recognisable (see for example Figure 1B). This is particularly true for endogenously tagged B56g, which is almost equally abundant at kinetochores and centromeres (see Figure 1—figure supplement 3D). At the very least, the partially overlapping localisation should be acknowledged.

We agree that it is not possible to stringently categorise localisation based on the results presented. The text has now been modified to make careful statements such as predominantly or mainly centromeric/kinetochore.

Kinetochore associated B56g, but not centromeric B56a, was sufficient to maintain the kinetochore functions of PP2A-B56 after the knockdown of all other B56 isoforms. The authors should explicitly state here that this cannot be explained by a lower expression level of B56a in these cells, as is only shown later in Figure 4A (but never commented upon).

A statement has been added to the text to explain that B56α and B56γ are both consistently expressed in HeLa cells (in relation to Figure 1—figure supplement 2 and Figure 1—figure supplement 3). Furthermore, the new data indicates that preserving B56α can only retain centromeric functions, whereas retaining B56γ only preserves kinetochore functions. This removes any concerns that the observed effects might be due to low level expression of one particular isoform.

The reverse experiment, addressing whether B56a alone, but not B56g, can maintain the centromeric functions of PP2A-B56 is lacking and the authors could elect to carry it out.

Indeed, we agree that this was a very important omission given that the title of the manuscript refers to shared labours between B56 isoforms at the kinetochore and centromere. New data has now been included to demonstrate that B56γ cannot support centromeric cohesion, whereas B56a can (Figure 1C).

The relative kinetochore intensity (e.g. Figure 1—figure supplement 5C) apparently refers to the combined kinetochore + centromere targeting. The normalised kinetochore intensity then shows the distribution between kinetochores and centromeres (e.g. Figure 1—figure supplement 5D). The y-axes should be re-defined to make this difference clear.

The axes and graph titles have now been changed to clarify this point.

Figure 1—figure supplement 2. Please include the WB for the other B56 isoforms.

A new western has now been inserted to include all B56 isoforms, except for B56β which was undetectable.

In Figure 2 and Figure 2—figure supplement 1, the authors aim to demonstrate that specific binding of B56a to Sgo2 contributes to its centromeric localization. With these figures, the following problems were identified:Figure 2C: Upon Sgo1 depletion, B56a seems to change its localization from a primarily centromeric localization to a primarily kt localization. Yet, this is not reflected in the quantifications shown in 2D and 2E.

The graph in Figure 2D shows relative intensity of centromere and kinetochore staining combined, which is why this does not change following Sgo1 depletion (B56a simply redistributes from centromere to kinetochore under this condition). The previous axis indicated “kinetochore intensities”, which was incorrect since the centromere/kinetochore cannot be distinguished with this type of analysis. All of these axes have therefore now been changed to read “Cen/Kt intensity” to avoid confusion.

The lineplots in Figure 2E allows the Cen/Kt can be visualised separately, and this demonstrates that B56a redistributes towards the kinetochore under this condition: the new Figure 2E now contains additional repeats to reduce variability and this does indeed demonstrate redistribution towards the kinetochore.

Furthermore, Figure 2D gives the impression that Sgo1 is very efficiently depleted in Sgo1RNAi cells. The IF image shown in C gives a different impression. Could the authors please comment on this?

We try to always choose images that most closely represent the mean data (a statement is included in the methods to indicates this). In the previous Figure 2D the punctate staining was actually background staining that do not consistently align with kinetochores and can also be seen in the siCon image. However, the punctate pattern visible in the zoom was indeed confusing since it did partially overlap with kinetochore. Therefore, a different region has now been selected to avoid confusion and to better represent the minimal kinetochore staining seen under this condition.

Subsection “Sgo2 provides specificity for centromeric B56 recruitment”: "…, CB-Sgo2 was only able to recruit B56a (Figure 2h-k)." This statement is puzzling. Figure 2J shows hardly any YFP-B56g localization. Why is the localization so different compared to Figure 1 and how was the quantification based on such IF images performed?

Thank you for pointing out this error. This image was not representative of the data because it had high background cytosolic levels and poor kinetochore staining. A new example image has now been included that better represents the mean data.

Importantly, it should be clarified whether this figure and the related Figure 2—figure supplement 1 are carried out under conditions of depletion of Sgo1 and/or Sgo2, as this does not seem to be stated anywhere.

The CB-Sgo1/2 experiments were all performed in the presence of endogenous Sgo1/2. This has now been clarified in the Materials and methods section.

Subsection “Sgo2 provides specificity for centromeric B56 recruitment”: "…CB-Sgo1 was able to localize both B56a and B56g to the inner kinetochore (Figure 2—figure supplement 1)." This statement is not supported by the data. Under wt conditions, B56g already colocalizes with CENP-C as shown in Figure 1A and 1B. Why are the quantifications (Figure 2K and Figure 2—figure supplement 1D) so different for control conditions?And given that B56g colocalizes with CENP-C under wt conditions, how do the authors come to the conclusion that CB-Sgo1 recruits B56g to inner kts?

Regarding points 3 and 4:

Under wt conditions, B56a and B56g already colocalise with the centromere or kinetochore, respectively (Figure 1A and B). The later figures with CB-Sgo1/2 (Figures2k, S6d etc.) still contain this baseline Cen/Kt localisation, but we then quantify the additional localisation induced by CB-Sgo1/2 expression. The same strategy was previously used by Meppelink et al., 2015 to show recruitment of B56d to CB-Sgo1. Our data indicate that CB-Sgo2 increases B56a (Figure 2I), but not B56g (Figure 2K), whereas CB-Sgo1 causes a large increase in both B56a and B56g at the Cen/Kt (Figure 2—figure supplement 1). Therefore, although the background Cen/Kt does complicate the overall interpretation (this is now clarified in the text), we still believe that our previous statements were supported by the data; given that we observe statistically significant increases only in certain conditions (i.e. in all CB-Sgo1 or -2 conditions, except for CB-Sgo2 + B56g). Please note, that we chose to use the inner kinetochore for these experiments in spite of these difficulties, because it may be important given that Aurora B activity was previously reported to be required for Sgo2:B56 interaction (Tanno et al., 2010).

Along the same line, B56a shows a stronger co-localization with CENP-C under control conditions than B56g (compare Figure 2—figure supplement 1B to Figure 2—figure supplement 1D).

Thank you for pointing this out. The image in S6D did not demonstrate the clear kt localisation in B56g controls. This image has now been changed for a more representative alternative (now Figure 2—figure supplement 1B and D).

The figure legend of Figure 2—figure supplement 1 (CB-Sgo1 recruits both B56a and B56g to centromeres) may be wrong. Apparently, what is meant is.….to inner Kts.

As stated above, we have now correctly labelled all these axes to indicate “Cen/KT intensities” since we cannot distinguish between these two localisations in this type of analysis.

Figure 3—figure supplement 2: There is hardly any signal for wt B56g (Figure 3—figure supplement 2A) that colocalizes with CENP-C. Could the authors please comment on this?

The cells imaged in Figure 3—figure supplement 2A are in interphase. Although B56γ is present in the nucleus in interphase, its main kinetochore receptor, BubR1, is not.: BUBR1 is only recruited to the kinetochore following nuclear envelope breakdown (see Nijenhuis et al., 2014). As such, in the wt situation prior to NEB, B56γ does not colocalize with CENP-C at kinetochores.

One of the reviewers raised a concern that the data are presented in a very compact manner that may not be readily accessible to a broad readership. While no specific suggestion was provided, the authors should consider this point and consider changes that will improve readability and raise the paper's general interest.

We thank the reviewer for this comment. We have now expanded the text slightly in key areas to explain things better to readers who do not have experience in kinetochore signalling. Hopefully this will now help to raise interest from a general audience.